# Precipitation Pattern in the Western Himalayas revealed by Four Datasets

Hong Li[1,2], Jan Erik Haugen[3], and Chong-Yu Xu[2]

[1]Norwegian Water Resources and Energy Directorate, Norway
[2]University of Oslo, Norway
[3]Norwegian Meteorological Institute, Norway

*Correspondence to:* Hong Li (lihong2291@gmail.com)

**Abstract.** Data scarcity is the biggest problem for scientific research related to hydrology and climate studies in the Great Himalayas Region. High quality precipitation data are difficult to obtain due to sparse network, cold climate and high heterogeneity in topography. In this paper, we examine four datasets in Northern India of the Western Himalayas: interpolated gridded data based on gauge observations (IMD, $1°\times1°$ and APHRODITE, $0.25°\times0.25°$), reanalysis data (ERA-interim, $0.75°\times0.75°$) and high resolution simulation by a regional climate model (WRF, $0.15°\times0.15°$). The four datasets show a similar spatial pattern and temporal variation during the period 1981-2007, though the absolute values vary significantly (497-819 mm/year). The differences are particularly large in July and August at the windward slopes and high elevation areas. Overall, the datasets show that the summer is getting wetter and the winter is getting drier, though most of the trends in monthly precipitation are not significant. Trend analysis of summer and winter precipitation at every grids confirms the changes. Wetter summers will result in more and bigger floods at the downstream areas. Warmer and drier winters will result in less glaciers accumulation. All the datasets show consistency in the period 1981-2007 and can give a spatial overview of the precipitation in the region. Comparing with the Bhuntar gauge data, the WRF dataset gives the best estimates of extreme precipitation. To conclude, we recommend the APHRODITE dataset and the WRF dataset for hydrological studies for their improved spatial variation which match the scale of hydrological processes as well as accuracy in extreme precipitation for flood simulation.

## 1 Introduction

The Great Himalayas Region is the largest cryosphere outside the polar areas and the source of many rivers which supply water to more than 800 million people (Hegdahl et al., 2016; Li et al., 2015b). The local population depends mainly on rivers for drinking water, hygiene, industry, fishing, but also for hydro-power generation and agriculture, which is one of main sectors of local economy (Ménégoz et al., 2013). Therefore, precipitation is very important to the local society and welfare of the local people. Climate change has significant impacts on water security, where mitigation and adaption to climate change are more challenging in this area due to poverty.

Precipitation is one of the most important elements in meteorology and hydrology. Precipitation measurements at gauges are usually used as benchmark data to compare with other datasets. They are often believed to be the most reliable and accurate

data. However, there are fewer gauges available in this area compared to other areas in the world. Therefore, it is tricky to look at spatial variability based on gauge data. Besides, quality of measurements is rarely high due to harsh climate and complex environment. Additionally, manual errors are very common in developing countries. These errors include, for example, error in gauge location, missing the unit of data as well as wrong position of the decimal point. Last but not least, gauge data are usually hard to obtain due to data policy and political conflict in some countries.

In recent years, with development of space-borne measurements and computing technologies, gridded precipitation datasets have been widely generated and attract much interest. Compared to measurements at traditional gauge, gridded data can cover a large area, sometimes even the globe, and disclose spatial variability at a continuous surface. Additionally, gridded data are usually produced by researchers for scientific purposes and they are free accessible to scientific research. Therefore, gridded data have been extensively used, particularly where high quality *in situ* measurements are not available.

There have been quite a few studies about precipitation over the Great Himalayas Region (Yatagai et al., 2012; Ménégoz et al., 2013; Palazzi et al., 2013). The available gridded data fall into four types: satellite data, interpolation of gauge observations, reanalysis and model simulation. However, all estimates are generally very uncertain due to the complex climate dynamics and local topography, and precipitation rates differ widely among the four types, even among different products of the same type. The satellite images show discrepancies due to platforms and characteristics of sensors. Reflectance from land surface, particularly snow and ice, can cause distinctive biases (Yin et al., 2008). The interpolated observations are usually believed the most reliable. However, great cautions have to be paid when using such data due to inadequacy of interpolating methods and unavoidable inferiors inherited from gauge measurements. For example, underestimation of precipitation could be 58% of annual total precipitation in the cold Alaska region due to wind, wetting loss and trace precipitation (Yang et al., 1998). High resolution climate models provide an alternative perspective and the models are competitive in aspect of high spatio-temporal resolution, identification of precipitation forms (Ménégoz et al., 2013), and internal consistency between climate parameters. On the other hand, the simulated data may misrepresent the reality and suffer from inadequacy of boundary and forcing conditions. Reanalysis data are a combination of observations from many sources and dynamic models, but users should be cautious because of continuous changes in observing systems and systematic model errors (Dee et al., 2011). Additionally, uncertainties in reanalysis data are difficult to understand and quantify (Dee et al., 2011). The weaknesses and strength of each type are summarized in Table 1.

In this study, we select four datasets from various sources, i.e. interpolation of gauge observations, reanalysis and model simulations in Northern India of the Western Himalaya as well as measurements at one rain gauge. Due to differences in availability, a common analysis is based on daily data in a long period of 27 years (1981-2007). To our knowledge, this is the first of its kind in this region in aspect of number of datasets and data length. The purpose is to compare the datasets and to find their similarity and difference, as well as implications for further use in hydrological studies.

## 2   Study Area

The study area lies in the western part of the Indian Himalayan Region (Figure 1). The highest point is 7677 meters above sea level (m a.s.l.), located in the north-eastern region. The low elevation part lies in the south-western region, which adjoins Pakistan. The climate is affected by monsoon and western disturbance. In summer, warm moisture from the Indian Ocean moves northwards and turns westward when it hits the high mountains. This interaction brings plenty of precipitation and daily precipitation can be more than 200 mm (Purohit and Kau, 2016). Precipitation at high mountains usually falls as snow in winter. Along the course of the moist wind, precipitation decreases from east to west. In winter, the climate is controlled by western turbulence. The mid-latitude low pressure systems bring some snowfall (Ménégoz et al., 2013), but winter is generally quite dry, especially in the coldest region. In this study, seasons are referred based on northern meteorological seasons (spring: March to May; summer: June to August; autumn: September to November; winter: December to February).

This area is the headwater of the Indus River and the Ganges River, which are transboundary among China, India, Pakistan and Bangladesh. Additionally, these two rivers have very high hydropower potential. How to explore hydropower is continuously negotiated among the involved countries, which makes the study area very political sensitive.

## 3   Data

### 3.1   IMD dataset

The IMD dataset is produced by the India Meteorological Department for the whole India. The time period is 1951-2007 and the spatial resolution is $1° \times 1°$. The data are interpolated from gauge measurements by using the Shepard method (Shepard, 1968). Rajeevan et al. (2006) compare the IMD dataset with the Variability Analysis of Surface Climate Observations (VASClimo) dataset and conclude that the IMD dataset is more accurate in terms of spatial variation. The IMD dataset has been extensively used in climate related research and applications, such as validation of climate models (Bollasina et al., 2011; Wiltshire, 2014) and monsoon variability and predictions (Goswami et al., 2006).

The number of used gauges varies during the period as well as spatially across the region. The average number of gauges per grid point is 2.99 ranging from 0.2 to 4.4 (Rajeevan et al., 2006). Spatially, more gauges are used in the central south; less gauges near the borders of India and in the northern part. No gauge measurements are available near the latitude of 35.5°N and northward.

### 3.2   APHRODITE dataset

The APHRODITE (Asian Precipitation—Highly Resolved Observational Data Integration Towards Evaluation of Water Resources) dataset is interpolated by the Sphere map method based on data collected at 5,000–12,000 gauges (Yatagai et al., 2012). The interpolated parameter is the precipitation anomaly or ratio, instead of the precipitation amount (Yatagai et al., 2012) . Ele-

vation corrections are considered by a weighting function, which is based on the angular distance with considering topography (Yatagai et al., 2012). The dataset covers Asia over the period of 1951-2007. Different versions of the APHRODITE dataset have been used to determine of Asian monsoon precipitation change, hydrological modelling (Pechlivanidis and Arheimer, 2015; Xu et al., 2016), verification of high-resolution model simulations and satellite precipitation estimates (Kamiguchi et al., 2010). In this research, we use the latest version (V1101) for monsoon Asia at a spatial resolution of $0.25° \times 0.25°$ (Dimri et al., 2013). The APHRODITE dataset uses the largest number of gauge observations among interpolated products, and is believed to be one of the most realistic precipitation datasets for Asia (Ménégoz et al., 2013).

### 3.3   ERA-interim dataset

The ERA-interim dataset is the precipitation product of the ERA-Interim (Dee et al., 2011), which is a spatially and temporally complete dataset of multiple climate variables at high spatial and temporal resolution. The data we use here are on a Gaussian grid (with a resolution of $0.7 \times 0.7°$ at the Equator) with a 3-hour time resolution, and aggregated to daily time step. The ERA-Interim is a global atmospheric reanalysis dataset produced by the ECMWF (European Center for Medium-Range Weather Forecasts) [1]. The dataset dates back to 1979 and is updated with approximately one month delay from real-time. The data assimilation system is based on a 2006 release of the IFS (Cy31r2) (Dee et al., 2011). This dataset has been widely used as boundary and forcing conditions for regional climate models (Dimri et al., 2013; Katragkou et al., 2015).

### 3.4   WRF dataset

The WRF dataset is generated by using a regional climate model, the Weather Research & Forecasting Model (v3.7.1). The climate model is a limited-area, non-hydrostatic, primitive-equation model with multiple options for various physical parameterization schemes. The model has been used in climate simulation in Asia and other areas (Maussion et al., 2011; Li et al., 2016). Here we use Thompson scheme for microphysics, CAM for short and long wave radiation, the Noah Land-Surface scheme, Mellor-Yamada-Janjic TKE for planetary boundary-layer and Kain-Fritsch (new Eta) for convection. The model is forced by 6-hourly ERA-interim reanalysis data. To avoid error at boundary edges and to facilitate further hydrological modelling work, we setup the model at a very large domain (59-91°E, 9-46°N). The spatial resolution is around 16 km, where topography and land use are aggregated from data with accuracy of 10 meters. They are preprocessed by using the WRF Preprocessing System (WPS). We divide the atmosphere into 30 vertical layers with model top pressure 50 hPa. The height of the lowest model level varies between 15 and 27 meters depending on the surface pressure. The whole simulation period is from 1979 to 2007, and the period 1979-1980 is used as model spin up. Due to long model running time, we restart the model around every five years. Model setup is summarized in Table 2 and the whole setting is a supplementary file.

---

[1] The next generation reanalysis, ERA5, featuring a higher horizontal resolution (~31 km) and a 10-member ensemble approach for uncertainty estimates, is released by the end of 2017. See e.g. http://www.ecmwf.int/en/newsletter/147/news/era5-reanalysis-production

### 3.5 Gauge data

The rain gauge Bhuntar lies in a valley at a small town in the state of Himachal Pradesh, India (Figure 1). The Bhuntar gauge is only 400 meters down the confluence of the Parvati River with the Beas River. The altitude of the gauge is 1080 m a.s.l., and both precipitation and discharge data are used in this study. Annual precipitation is 921 mm/year based on data from 1981 to 2007 with most rain falls in July and August. Temperature is rarely below 0°C , and only minimum temperature is occasionally below 0°C in winters. The precipitation data have been used in hydrological modelling research for the Beas Basin (Li et al., 2015a).

### 3.6 Discharge

Three discharge series are selected to cross validate water balance. They are respectively Pandoh (downstream), Bhuntar (middle stream) and Manali (upstream). These stations are operated by Central Water Commission regional office in India. The catchments are located in the Beas basin, which is a main tributary of the Indus River in Northern India (Figure 1). The catchments are nested from upstream to downstream. The purpose is to reflect precipitation data at various elevations within a hydrological scale. Runoff is considerably influenced from glacier melting (Li et al., 2015a). According to the 0.5 km MODIS-based Global Land Cover Climatology by the USGS Land Cover Institute (https://landcover.usgs.gov/global_climatology.php), coverage of snow and ice is 16% in the Pandoh catchment, 24% in the Bhuntar catchment and 21% in the Manali catchment. The discharge data have been manually quality controlled and missing data are filled by discharge anomaly. Discharge measurements are more qualified than precipitation in the snow and ice dominated area (Henn et al., 2015; Kretzschmar et al., 2016). Therefore, the quality of runoff simulation can infer by the forcing precipitation data. Li et al. (2016) use the WRF-Hydro (v3.5.1) modeling system in the Beas Basin, and they find that the distribution of simulated daily discharge values agrees well with observations, which reversely confirms the precipitation simulations.

### 3.7 Evaporation

The MODIS Global Evapotranspiration Project (MOD16) (http://www.ntsg.umt.edu/project/modis/mod16.php) is selected to reveal actual evaporation. The MODIS project is started in 2000, and has a short overlap period with the study period. Additionally, part of the catchments is covered by permanent snow and ice and the sensors cannot work well on this type surface. Therefore, we use annual mean amounts of 2000 to 2013 to reduce uncertainties. The missing ratios of annual mean actual evaporation are 22% for the Pandoh catchment, 32% for the Bhuntar catchment and 31% for the Manali catchment.

## 4 Results

### 4.1 Spatial variations

The four datasets show similar spatial pattern of mean annual precipitation (Figure 2). The highest precipitation is located at the foothill of the mountains and stretched from southeast to northwest. Visually, the high precipitation belt (the foothills of the

mountains and the southeastern corner) is most clearly shown by the WRF dataset. The spatial variability increases from the IMD dataset to the WRF dataset. Their coefficients of variation are respectively 0.5 for the IMD data, 0.6 for the ERA-interim data, 0.7 for the APHRODITE data and 1.1 for the WRF data. The density curves of mean annual precipitation values in all grid points (Figure 3) and the statistics of the Kolmogorov–Smirnov test (Table 3) show the variabilities and the differences

among the datasets more clearly.

Both the IMD and APHRODITE datasets are interpolated from observations at gauges. However, the APHRODITE dataset shows a rain belt at the mountains' foothills much better. Additionally, the APHRODITE dataset shows much lower estimates (less than 300 mm/year) at the north-eastern corner. This area is quite high, with mean elevation at 4650 m a.s.l. and elevation

ranges from 906 to 7677 m a.s.l. The temperature is -2.35 °C of annual mean and as low as -16.81 °C in January (AphroTemp, Yatagai et al., 2012). The reason for this low precipitation area is that the APHRODITE dataset uses more gauges, particularly also observations from Nepal, Bhutan and China (Yatagai et al., 2012). These gauges have undercatch problems, which means rain gauges could only catch part of snowfall due to wind and disturbance. In contrast, the IMD dataset uses only the gauges at the low valley area of India and extends to north by interpolation (Rajeevan et al., 2006). Eventually, the APHRODITE dataset

has the lowest annual amount, only 61% of the IMD dataset.

The ERA-interim and WRF datasets are products with different dynamical models. The ERA-interim data and the WRF data are similar in terms of annual total amount (ERA-interim: 718 mm/year, WRF: 688 mm/year) and spatial pattern, partially due to the fact that in this area the observations that are assimilated into the data assimilation system are sparse and unevenly

distributed. The WRF data are more realistic than the ERA-interim data due to finer spatial resolution, especially in complex topography areas (Ménégoz et al., 2013; Dimri et al., 2013).

The effects of location and topography are shown in Figure 4. The summer precipitation changes dramatically. Over the high flat plateau, precipitation decreases with latitude since the strength of the monsoon decreases with distance from its source. As

the monsoon gets closer to the mountains, precipitation starts to increase. As the air parcel is lifted to high elevation, climate gets dry and cold. The winter precipitation occurs mainly along the upslope. The magnitude is also small and decreased along the path of the winter monsoon. The highest precipitation occurs in the windward of the upslope region, but it is 0.5 or 1.5 degree (around 55-110 km) far away from the mountains in summer. Bookhagen and Burbank (2006) analyze a decade of TRMM data and also find the highest annual precipitation is offset by a few 10s of km south of either high topography or relief.

This offset has been found only over tall and broad mountain regions rather than narrow mountain peaks (Dimri and Niyogi, 2013).

The differences among the datasets are more obvious in summer at the mountain foot. The WRF dataset gives much more precipitation (700 mm/month) in July and August at the mountain foot, almost two times of other datasets (300 mm/month).

This is reported as a moisture bias in summer (Srinivas et al., 2013; Li et al., 2016). It is often cited as orographic bias which

describes as strong over-prediction of precipitation rates along windward slopes while predicted snowfall lies under measured values along leeward slopes (Maussion et al., 2011).

## 4.2 Temporal variations and changes

The inter-annual patterns are very similar as indicated by high correlations between pairs of datasets, shown in Table 4. The correlation between the IMD and APHRODITE datasets is the highest, reaching 0.91. The WRF dataset has low correlation with all other datasets. Spatially, the four datasets show a similar seasonal distribution, and the WRF dataset has the highest variability (Figure 5). The intra-annual cycle is also similar as shown in Figure 6. The WRF and APHRODITE datasets have respectively the highest and lowest precipitation in summer.

To look at changes over time, we select the Theil-Sen median method to calculate trend due to its robustness and the non-parametric Mann-Kendall test for the significance test. The trend analysis and significance test are done for areal mean of each month (Figure 6), and every individual grid for summer (Figure 7 and 8) and winter (Figure 9 and 10). The figures show increase in summer precipitation and decrease in winter precipitation, although both increase and decrease exist in each dataset. Three of the areal mean trends (May by the WRF dataset; June by the IMD and ERA-interim datasets) are statistically significant at the 95% confidence level. The spatial distribution of trends in summer precipitation varies a lot. Most decreasing trends of winter precipitation occurs at the northern part. Approximately 10% grids are significant at the 10% confidence level.

It is difficult to conclude why Northern India of the Western Himalayas shows an increase in summer precipitation. However, Bollasina et al. (2011) finds the same increasing monsoon precipitation in Northern India, whereas decreasing in the Central Asia. They use a series of climate model experiments, and conclude that such pattern is a robust outcome of a slowdown of the tropical meridional overturning circulation, which could be attributed mainly to human-influenced aerosol emissions. The trends will continue and become more significant with time if greenhouse gas emission continues as usual. Such trends would lead to strong negative mass balance conditions of glaciers, which is discussed in the next section.

## 5 Discussions

### 5.1 Comparison gridded precipitation datasets with gauge data

To compare the gridded datasets with measurements at the Bhuntar gauge, we extract the time series at the nearest point to the Bhuntar gauge. We look at annual precipitation (Figure 11), monthly anomaly (Figure 12) as well as extreme precipitation, i.e. annual maximum daily precipitation (Table 5). As shown in Figure 11, all gridded datasets are comparable with the Bhuntar gauge data. The interpolated datasets, IMD and APHRODITE are quantitatively closest to the Bhuntar gauge data. The ERA-interim and WRF datasets generally give two or three times higher precipitation than the Bhuntar gauge data. Figure 12 shows the differences are mainly from March to July in the WRF dataset, and July and August in the ERA-interim dataset. In addition,

the WRF dataset shows large variations from February to June, and the ERA-interim dataset shows large variations in July and August. Table 5 shows the statistics of annual maximum daily precipitation. Notably, the WRF dataset gives the closest estimate to the Bhuntar data in five quantiles, and the APHRODITE dataset gives the best estimate of the maximum precipitation over the whole period.

## 5.2   Comparison gridded precipitation datasets with runoff data

The annual actual evaporation from MODIS data is 614 mm/year at the Pandoh catchment, 639 mm/year at the Bhuntar catchment and 649 mm/year at the Manali catchment. The values are too high compared with 64mm/year at the Pandoh catchment for the period from 1990 to 2004 calculated by Kumar et al. (2007) using potential evaporation, mean and maximum temperature. The Pandoh catchement covers the lower and middle parts, and should have the highest evaporation due to warm climate among three catchments. The MODIS data are not qualified at the catchments and at small catchment scales for the study period.

The precipitation and runoff relationship is shown in Figure 13 as accumulation of monthly precipitation and runoff. Though the lines have different slopes, but they share very similar linear relationships. They are consistent in terms of temporal changes. Errors are systematic within each dataset. Runoff is generally less than precipitation due to evaporation loss. However, runoff could possibly exceed precipitation at glacierised catchment due to glacier melting. Runoff is most more than precipitation in the Manali catchment. In the Bhuntar catchment, only the ERA-interim data show less runoff than precipitation. All datasets show less runoff than precipitation in the Pandoh catchment. Precipitation is definitely underestimated at higher elevation area, especially in the Manali catchment. Azam et al. (2014) reconstruct annual mass balance of Chhota Shigri glacier since 1969. The Chhota Shigri glacier lies in the Western Himalaya, India and it is representative in terms of mass balance for the Western Himalayas glaciers (Azam et al., 2014). The mass loss rates are 0.36±0.36 for 1969 to 1985 and 0.57±0.36 m water equivalent per year (m w.e.a$^{-1}$) for 2001 to 2015. The runoff contribution from glacier melting is only 3306 mm within 29 years with assumptions of 20% glacier coverage and -0.57 m w.e.a$^{-1}$.

## 5.3   Implications for glaciers

In the Great Himalayas Region, there are many glaciers and they are key indicators of regional climate change and water resources. Temperature in combination with precipitation controls survival of glaciers. Therefore, we also look at changes in temperature by comparing the temperature results by the same simulation of the WRF precipitation dataset for the first and last five years, namely 1981-1985 and 2003-2007. We skip the trend analysis and significance test, because it is already well known that temperature is increasing fast in the Great Himalayas Region since 1980s (Ren et al., 2017). Temperature is well measured and simulated. Therefore, there is no need to go through many datasets. We are particularly interested in temperature at the equilibrium line altitude (ELA). As the slope of the regression lines shown in Figure 14, the WRF model is able to reproduce the lapse rates. Between the two five-years, temperature increases 0.91°C in winter and 0.26°C in summer. Such changes lead to an increase in the elevation of the freezing point (0°C) of 125 m in winter and 32 m in summer. As shown in Section 4.2, precipitation overall decreases in winter. In combination with increasing temperature, this is an unfavourable condition for

the glaciers with less accumulation and faster melting. Moreover, the area between 4900 m a.s.l., which is the equilibrium line altitude (ELA) of the Chhota Shigri glacier (see Azam et al., 2012, Figure 2), and 5200 m a.s.l. is large. Therefore, as the climate gets warmer, the ELA will further move up. Such a nonlinear characteristic of elevation distribution results in a potential large reduction in the accumulation area and small storage buffer of permanent snow and ice.

## 6   Conclusions

Data scarcity is a major problem for hydrological research in the Great Himalayas Region. High quality precipitation data are difficult to obtain due to the sparse network, cold climate and high heterogeneity in topography. This paper investigates the spatial and temporal pattern of precipitation in this region based on four datasets: interpolated gridded data based on gauge observations (IMD, $1° \times 1°$ and APHRODITE, $0.25° \times 0.25°$), reanalysis data (ERA-interim, $0.75° \times 0.75°$) and high resolution simulation by a regional climate model (WRF, $0.15° \times 0.15°$) in Northern India of the Western Himalayas during the period 1981-2007.

The four datasets are similar in terms of spatial pattern and temporal variation and changes, though the absolute values vary a lot (497-819 mm/year) due to the data source and the methods of data generation. The differences are particularly large in July and August and at the windward slopes and the high elevation areas. The datasets reveal that summer gets wetter and winter gets drier, though most of the trends are not statistically significant. Wetter summer results in more and bigger floods at the downstream areas. Warmer and drier winter results in less glaciers accumulation. The four datasets are able to give a good overview of spatial pattern and temporal changes. Comparison with measurements at the Bhuntar gauge shows that the WRF and APHRODITE datasets give the best estimate of extreme precipitation amounts. To conclude, the APHRODITE and WRF datasets are recommended for hydrological studies due to their improved spatial variations which match the scale of hydrological processes as well as accuracy in extreme precipitation for flood simulation. However, careful local correction is definitely required.

*Acknowledgements.*  This study is funded by the Research Council of Norway through the research program NORKLIMA under grant the project 216546. We thank India Meteorological Department and Sonia Grover at Water Resources Division at TERI (India), European Centre for Medium-Range Weather Forecasts and the APHRODITE (http://www.chikyu.ac.jp/precip/english/products.html) and the research program JOINTINDNOR under grant the project 203867 for provision of data. We thank Oskar Landgren at the Norwegian Meteorological Institute for help in modelling and data analysis as well as review of the manuscript. The model simulation was done when the first author worked at the Norwegian Meteorological Institute.

### Conflict of Interest

No conflict of interest.

## 7    Code availability

## 8    Data availability

The ERA-interim and APRODITE data are available from the data provider sites. The WRF data are available via https: //archive.norstore.no/pages/public/about.jsf.

5    *Author contributions.*  Hong Li: model simulation, data analysis and writing. Jan Erik Haugen: model simulation, analysis of results, review and revision. Chong-Yu Xu: review and revision.

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

**Figures**

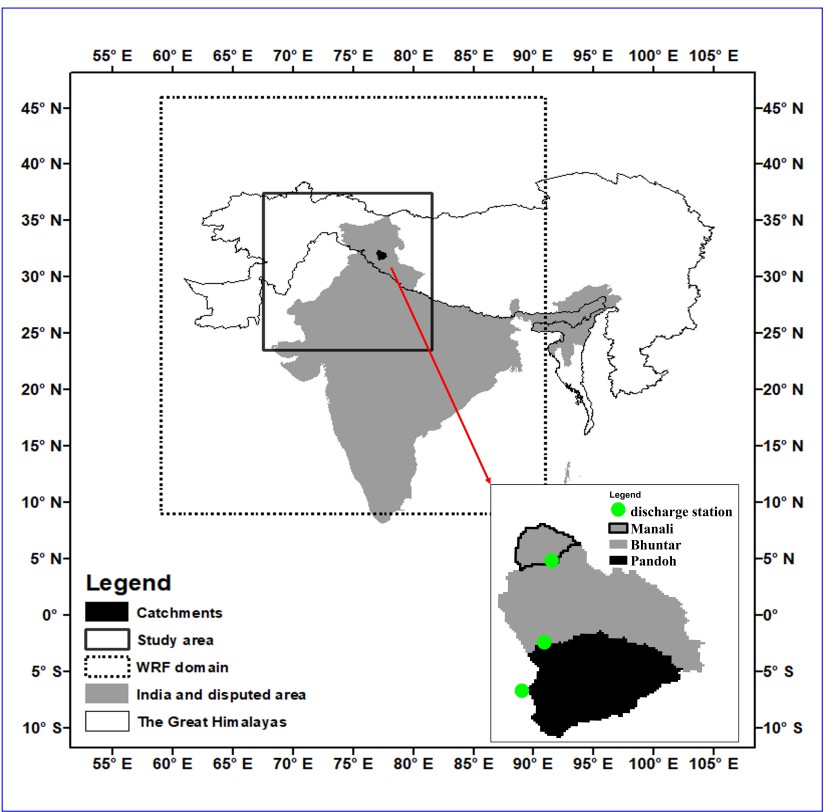

**Figure 1.** Location map of the study area, the Bhuntar rain gauge and three discharge stations.

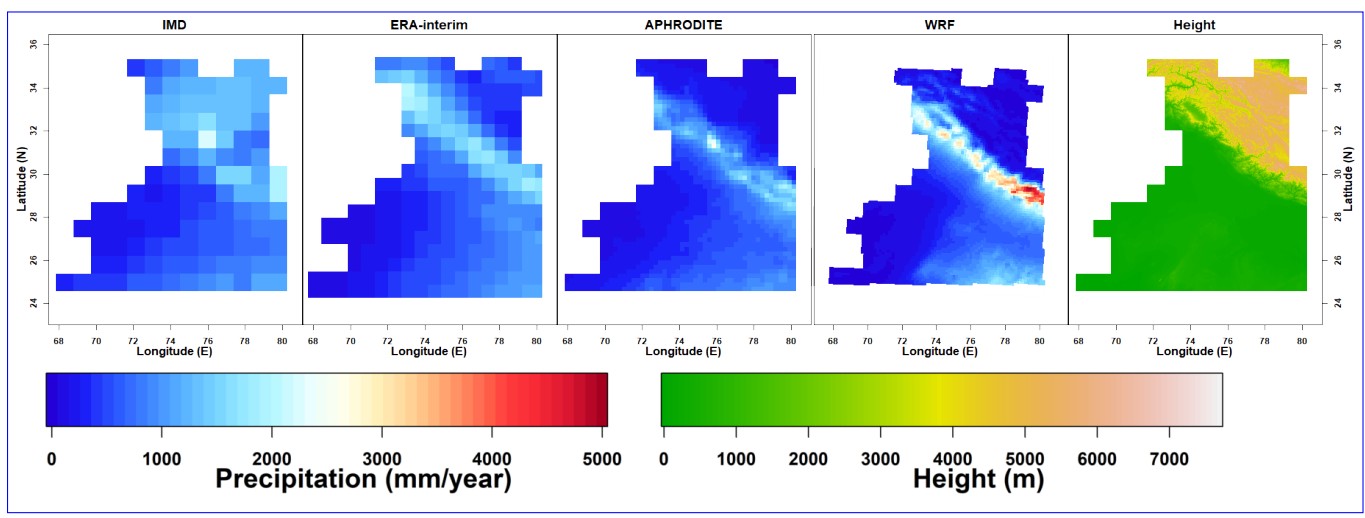

**Figure 2.** Mean annual precipitation (1981-2007) of the four datasets (from left IMD gridded observations, ERA-interim reanalysis, APHRODITE gridded observations and WRF regional climate model simulation) and terrain height of the study area(right).

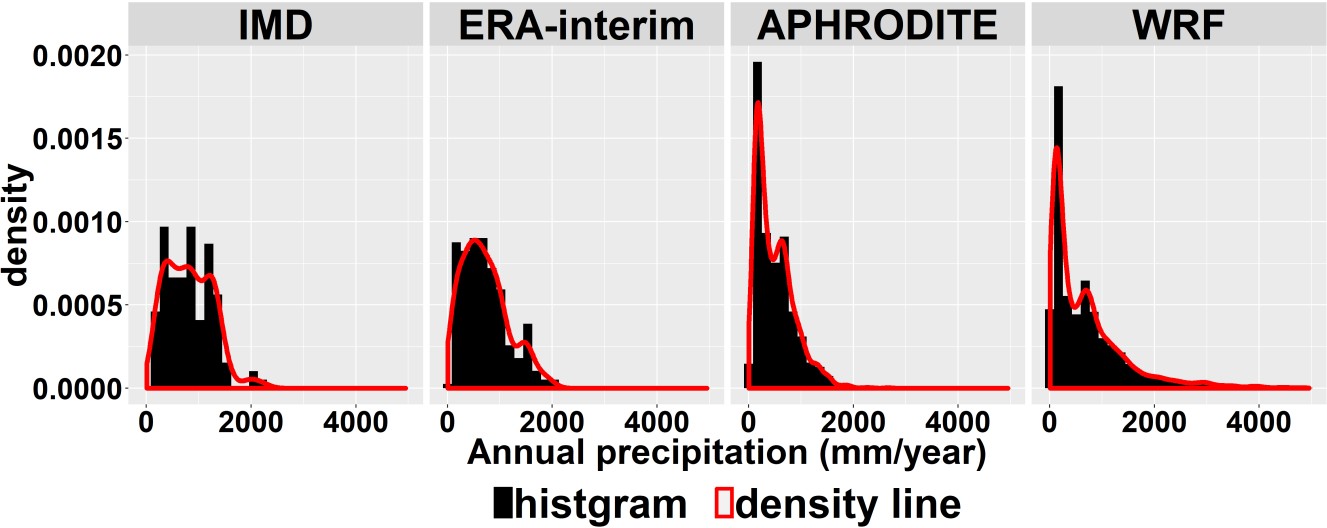

**Figure 3.** Density curves of mean annual precipitation values in all grid points.

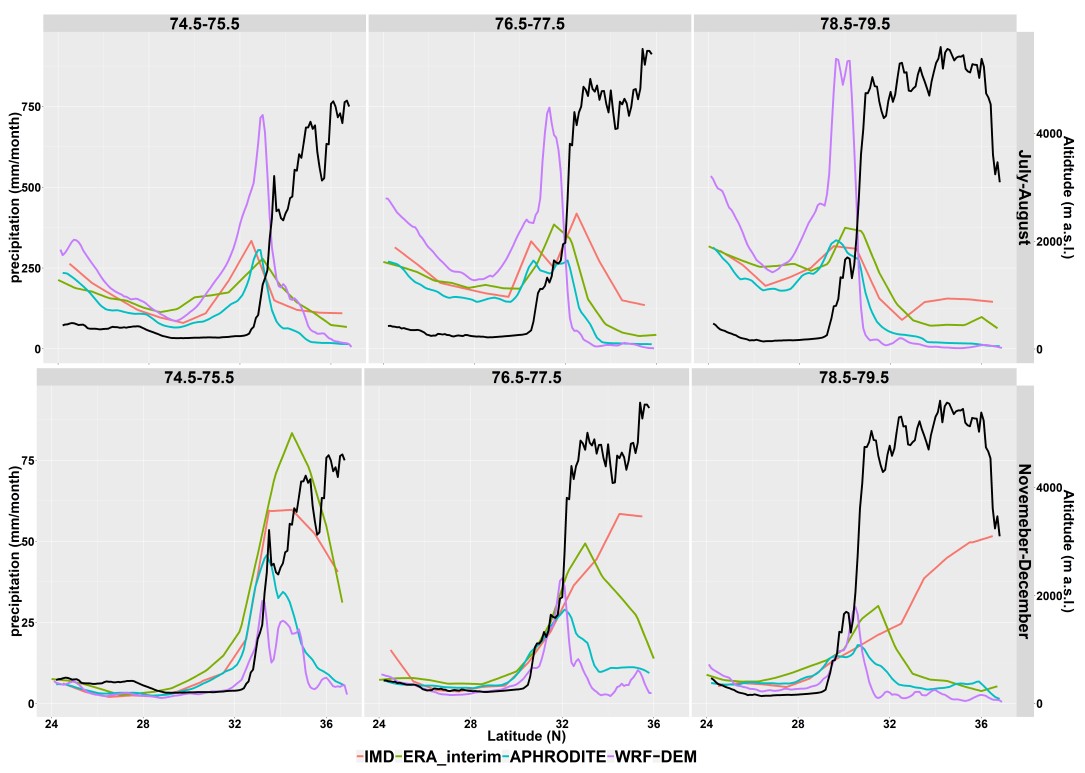

**Figure 4.** Precipitation (mm/month) for July-August (upper) and November-December (lower) and in three selected longitude bands from west to east (left to right) plotted against latitude. The longitude value of each band is indicated above the figures. The corresponding terrain height (m) in black is displays at the right axis.

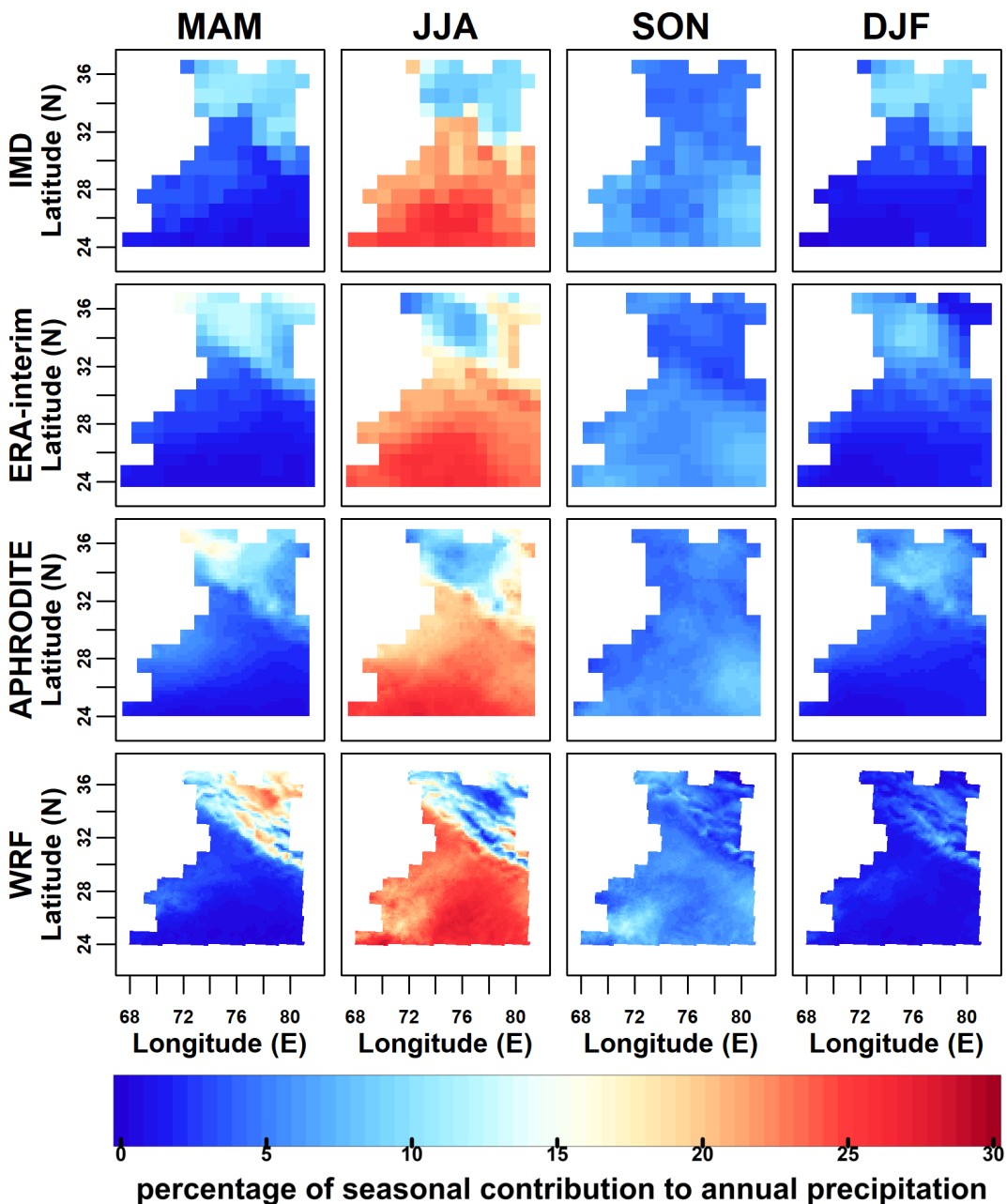

**Figure 5.** Seasonal contributions (in %) to annual precipitation. From left to right spring (MAM), summer (JJA), autumn (SON) and winter (DJF). From bottom to top WRF regional climate model, APHRODITE gridded observations, ERA-interim reanalysis and IMD gridded observations.

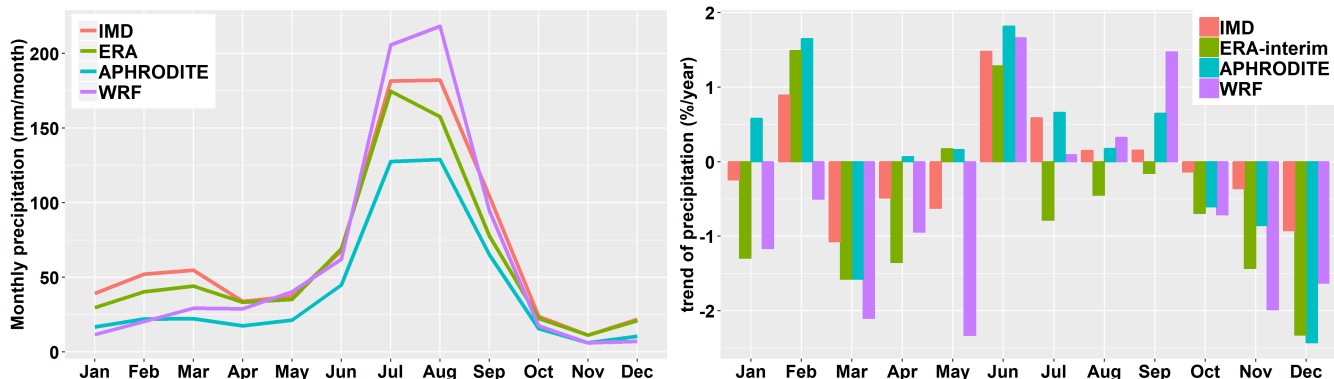

**Figure 6.** Monthly precipitation (left) and the trend during 1981-2007 (right).

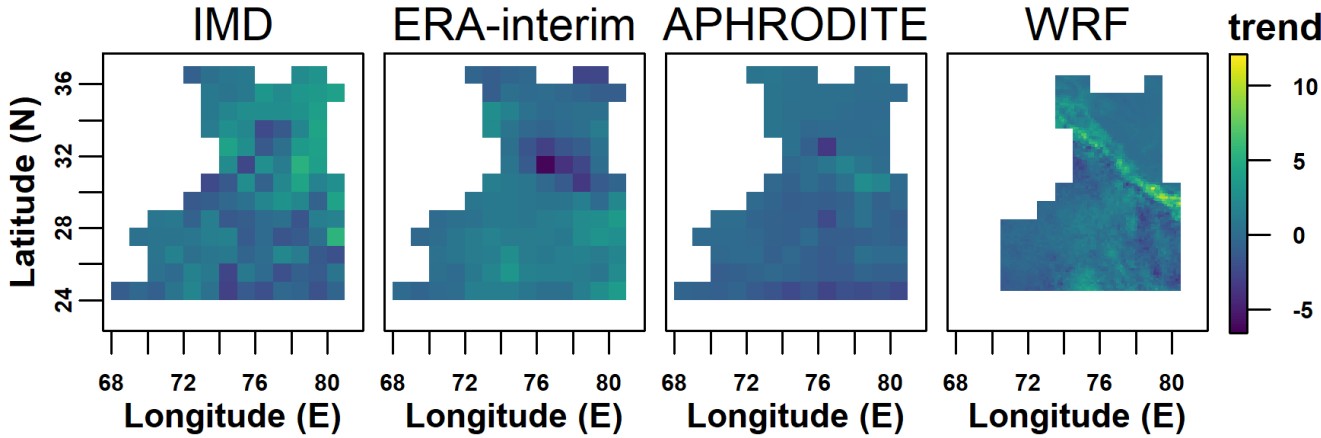

**Figure 7.** Trend (mm/month/year) of summer precipitation.

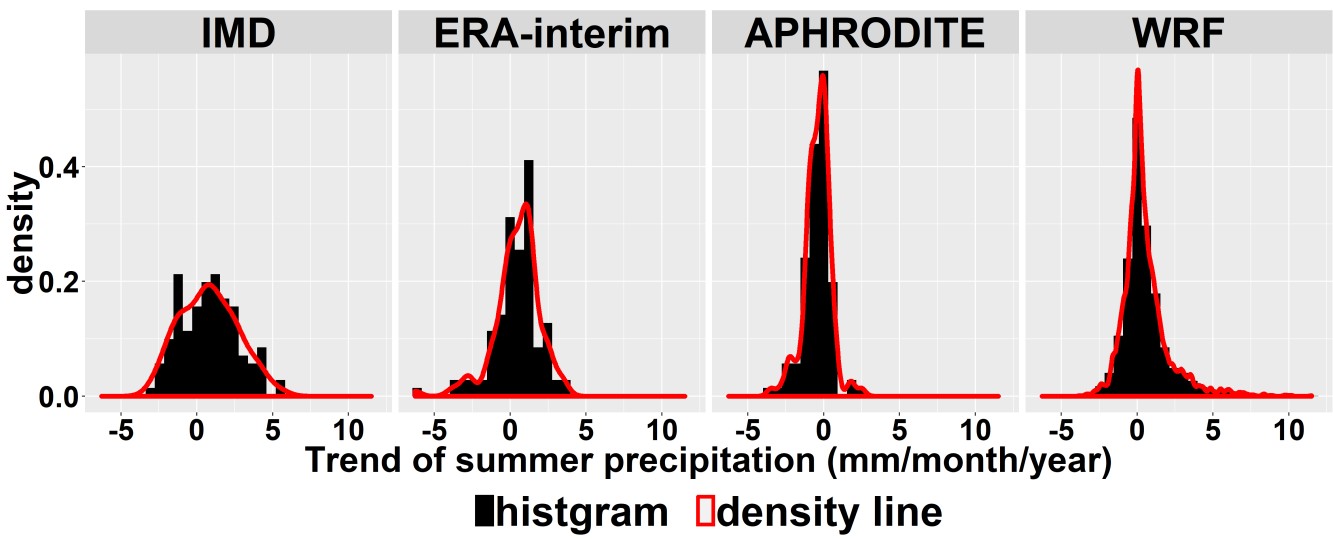

**Figure 8.** Density curves of trends (mm/month/year) of summer precipitation in all grid points.

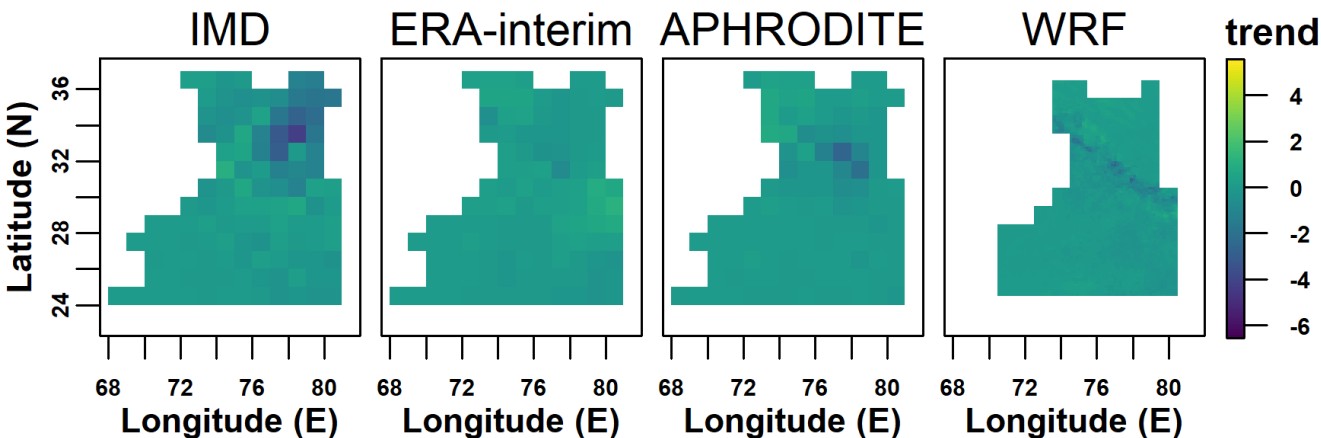

**Figure 9.** Trend (mm/month/year) of winter precipitation.

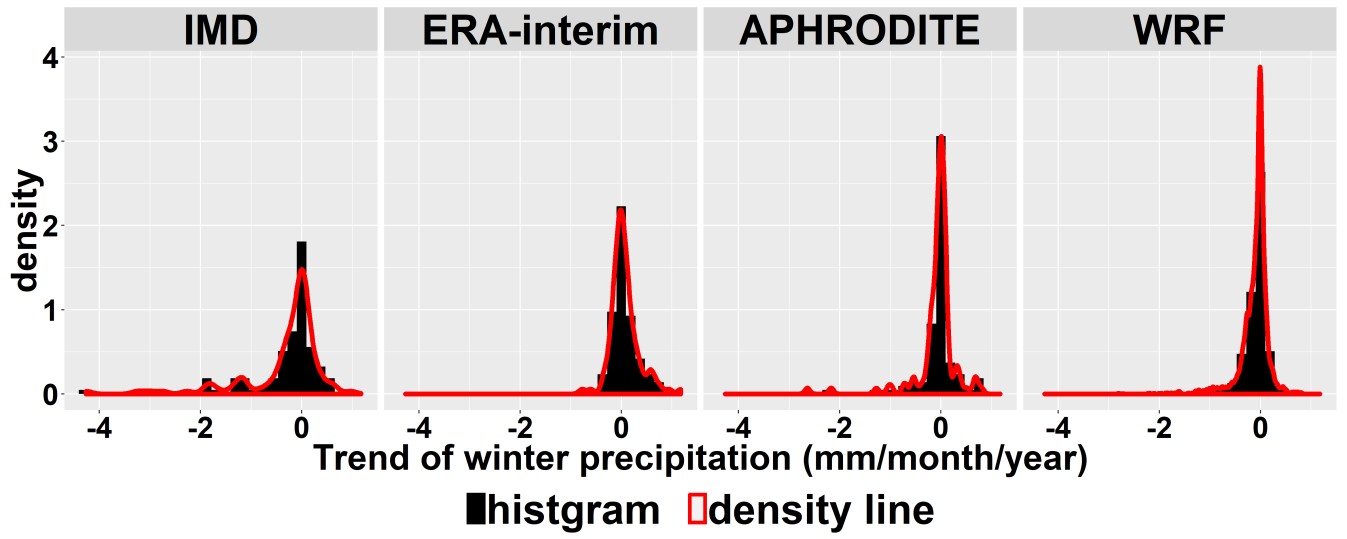

**Figure 10.** Density curves of trends (mm/month/year) of winter precipitation in all grid points.

**Figure 11.** Annual precipitation at the Bhuntar gauge. Data at the nearest point to the Bhuntar gauge are extracted from the gridded datasets

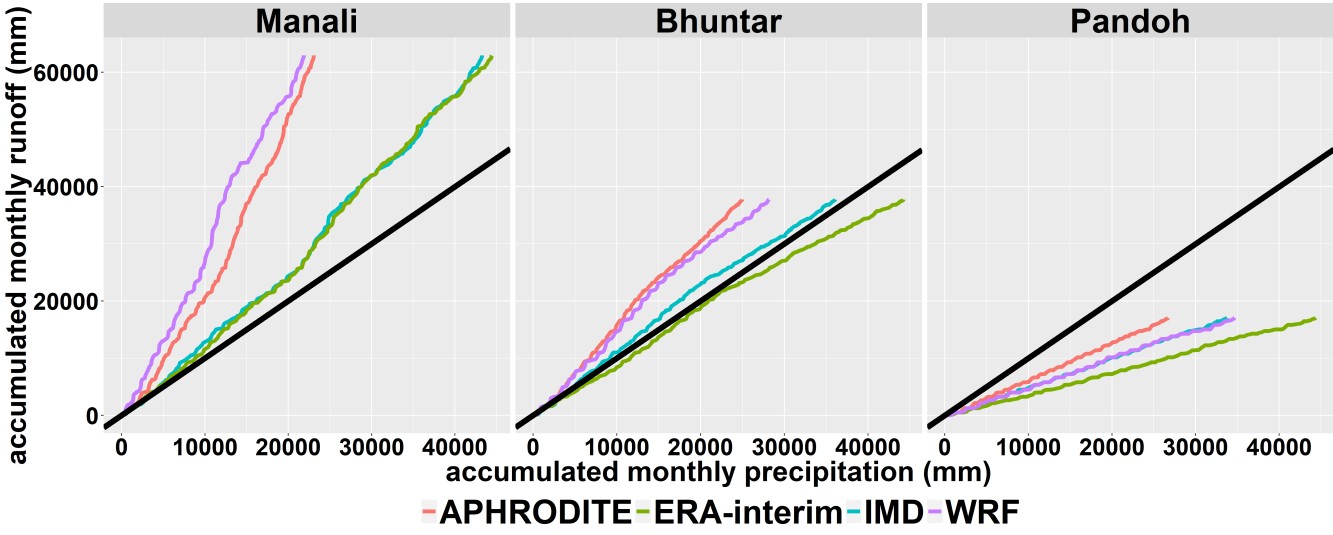

**Figure 12.** Monthly anomaly at the Bhuntar gauge. Data at the nearest point to the Bhuntar gauge are extracted from the gridded datasets.

**Figure 13.** Accumulated precipitation and discharge.

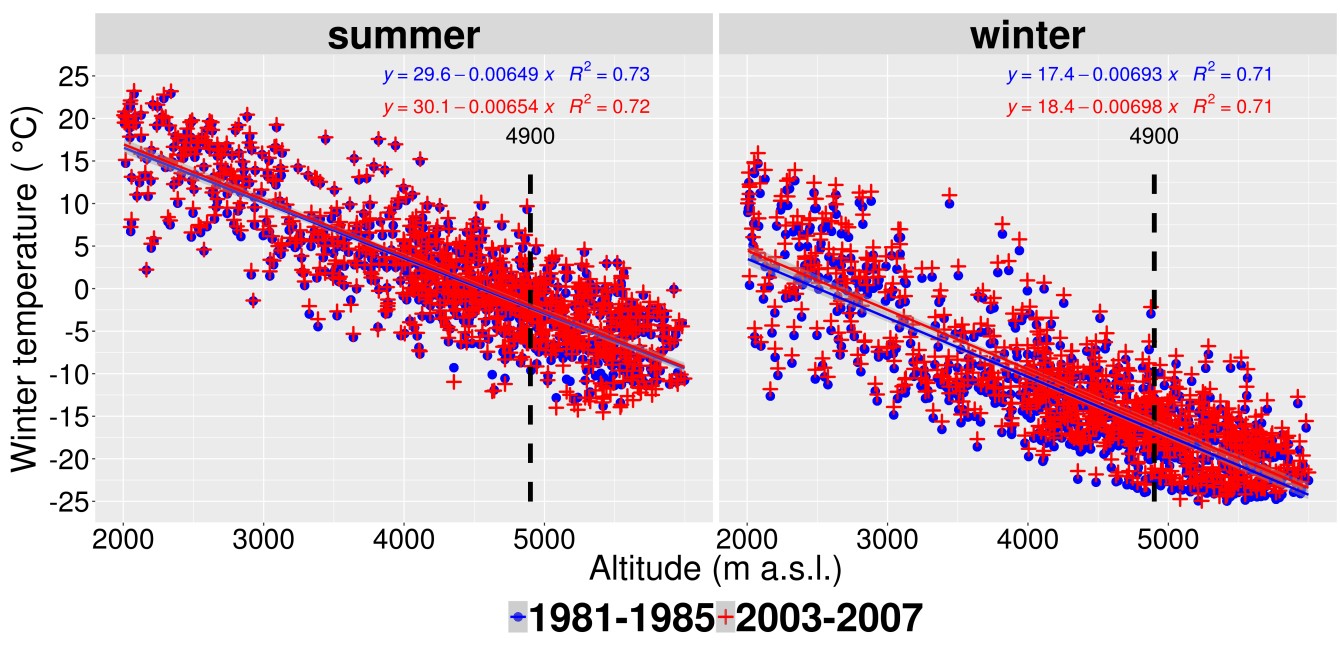

**Figure 14.** Mean temperature and its regression lines for the periods of 1981-1985 and 2003-2007 by the WRF simulation.

## Tables

**Table 1.** Summary of weaknesses and strength of four types of gridded precipitation data.

| Data type | Strength | Weakness |
|---|---|---|
| gauge | original ground measurements<br>long application | coarse distribution<br>undercatch of snow and rain due to wind<br>manual errors<br>high expense or unavailability due to political reasons |
| satellite | spatial observations<br>quality not affected by wind or other weather conditions | dependence on platforms and sensors<br>bias caused by snow and ice |
| interpolations | consistent with traditional ground observations | Inadequacy of interpolating methods<br>unavoidable inferiors inherited from gauge measurements |
| output from climate models | consistent with other meteorological parameters<br>possibility to measure uncertainties | inadequacy in algorithms, boundary and forcing |
| reanalysis | combination of modeling technique and many types of observations | changes in observation system<br>model error |

**Table 2.** The main settings of the WRF regional climate model. The complete setting are shown in supplementary file.

| Time and Domain | |
|---|---|
| Period | 1979-2007 |
| Region | 59-91 °E, 9-46 °N |
| Horizontal grid spacing | 16 km |
| Dimension | (193, 241, 30) |
| Model top pressure | 50 hPa |
| Height of the lowest level | 15-27 m |
| Physics | |
| Microphysics | Thompson scheme |
| Shortwave radiation | CAM |
| Longwave radiation | CAM |
| Surface-layer | Monin-Obukhov (Janjic) scheme |
| Land surface | Noah Land-Surface Mode |
| Planetary boundary layer | Mellor-Yamada-Janjic TKE scheme |
| Cumulus | Kain-Fritsch (new Eta) scheme |
| Lateral boundaries | |
| Forcing | ERA-Interim $0.75° \times 0.75°$, 6 hourly |

**Table 3.** P-value of the tailed Kolmogorov–Smirnov test on differences of on annual precipitation (mm/year) among the datasets. The p-value indicates strong evidence against the null hypothesis. It is typically to reject the null hypothesis, which is two datasets are the same here, when p-value is not greater than 0.05.

| Data \ Data | IMD | ERA-interim | APHRODITE | WRF |
|---|---|---|---|---|
| IMD | - | 5.4e-02 | 3.6e-11 | 3.7e-10 |
| ERA-interim | 5.4e-02 | - | 5.0e-11 | 2.5e-11 |
| APHRODITE | 3.6e-11 | 5.0e-11 | - | <2.2e-16 |
| WRF | 3.7e-10 | 2.5e-11 | <2.2e-16 | - |

**Table 4.** Pearson's correlation of annual precipitation series

| Data \ Data | IMD | ERA-interim | APHRODITE | WRF |
|---|---|---|---|---|
| IMD | - | 0.86 | 0.91 | *0.64* |
| ERA-interim | 0.86 | - | 0.86 | *0.64* |
| APHRODITE | 0.91 | 0.86 | - | *0.59* |
| WRF | *0.64* | *0.54* | *0.59* | - |

**Table 5.** Statistics of annual maximum daily precipitation (mm/day) at the Bhuntar gauge. Data of the nearest point are extracted from the gridded datasets. Bold and italics indicates the value closest to the data of the Bhuntar gauge.

| Data \ quantiles | minimum | 0.05 | 0.25 | median | 0.75 | 0.95 | maximum |
|---|---|---|---|---|---|---|---|
| Gauge | 38.0 | 41.1 | 57.8 | 69.6 | 82.2 | 104.3 | 106.0 |
| IMD | 26.4 | 30.2 | 36.1 | 52.1 | 67.3 | 116.8 | 147.3 |
| ERA | 59.2 | 67.7 | 80.2 | 94.0 | 121.7 | 149.5 | 154.4 |
| APHRO | 28.2 | 29.8 | 38.6 | 52.9 | 58.0 | 70.9 | ***103.1*** |
| WRF | ***43.5*** | ***51.0*** | ***57.3*** | ***65.7*** | ***78.3*** | ***93.5*** | 99.8 |

## Appendix A: Appendix

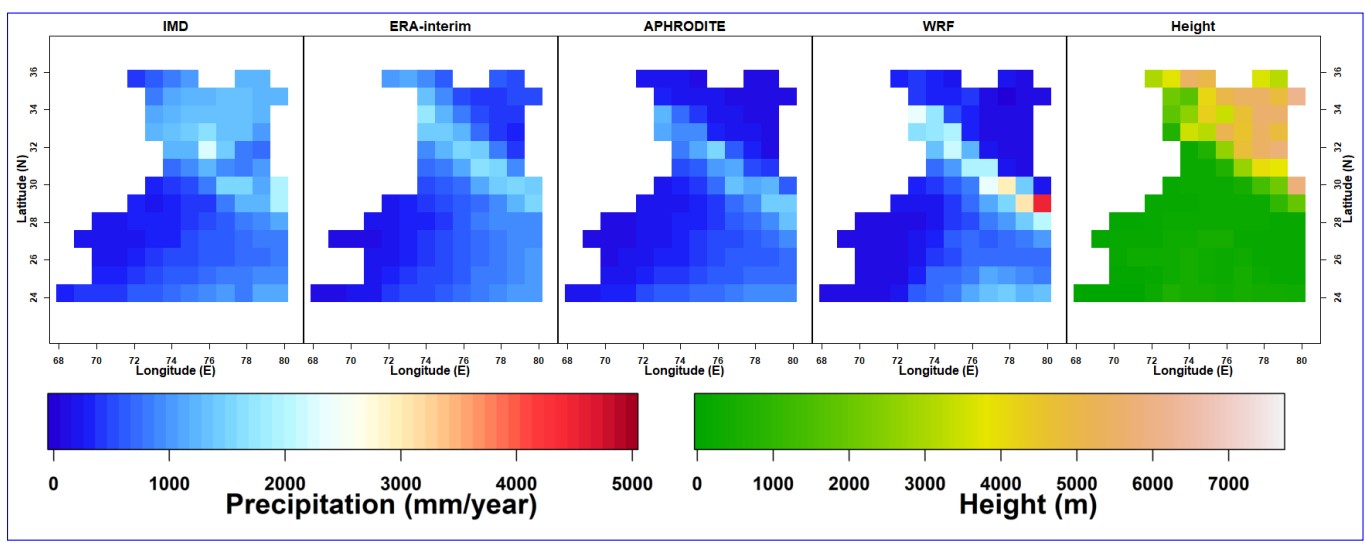

**Figure A1.** Precipitation of the gridded datasets and terrain height of the study area resampled to the IMD grid by bilinear interpolation.

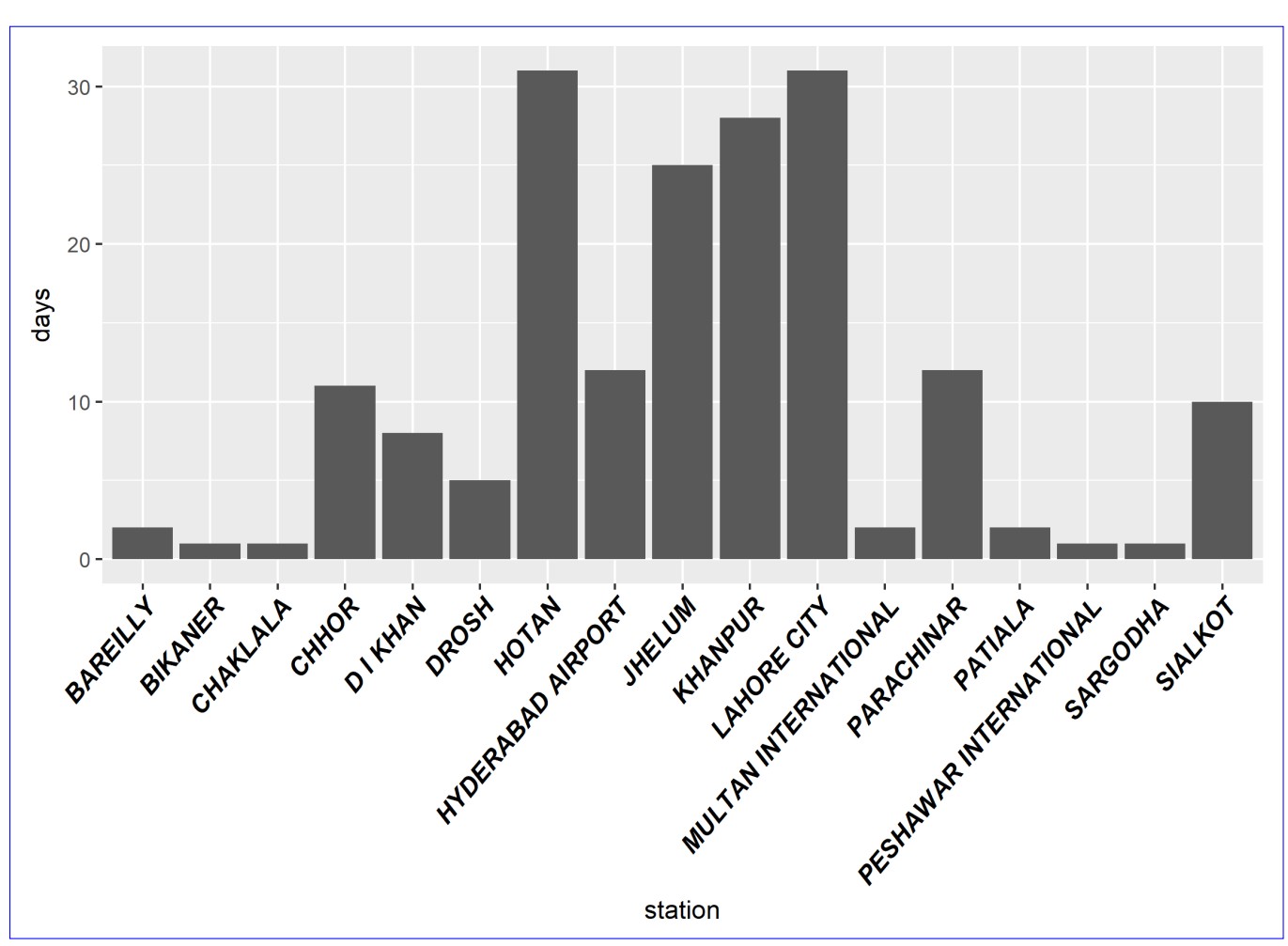

**Figure A2.** Data missing rate in the National Climatic Data Center in the study areas.

**Table A1.** Statistics of the non-parametric Mann-Kendell test of precipitation.

| Data | Trend (mm/year) | p-value |
|---|---|---|
| IMD | -2.01 | 0.509 |
| ERA-interim | 0.86 | 0.994 |
| APHRODITE | -3.59 | 0.098 |
| WRF | -0.70 | 0.819 |