# Peer review of "Precipitation Pattern in the Western Himalayas revealed by Four Datasets"

_Hydrology and Earth System Sciences, 2017_

## Referee Comment (RC1) · Anonymous Referee #1 · 4 Jul 2017

Major comment 1: The methodology of attempting to distinguish precipitation trends from the four types of dataset is not scientifically valid. Firstly, there is no attempt to calculate the inter-annual variability of precipitation for each of the datasets. Therefore, it is impossible to tell whether the changes between the 2003-2007 and 1981-1985 periods are meaningful. There was also no justification for why these periods were even chosen. It is also impossible to tell whether choosing five years for each period is large enough to capture the precipitation representative of the 1980's and 2000's, i.e. these periods could easily have been anomalous. There is also no physical justification given as to whether the e.g. increase in summer precipitation during these two periods is physically consistent with either dynamic or thermodynamic large-scale changes, such as in response to the observed weakening of summer monsoonal precipitation (e.g.

[Figure]

Bollasina et al. 2011). The differences also aren't quantified, and are referred to in the Abstract as 'an increase in summer and a decrease in winter with large variations'. The fact that there are large differences between the datasets is also a cause for concern, with the differences between datasets being possibly greater than the magnitude of the differences between the two periods. Secondly, the authors showed that the linear precipitation trend was insignificant, which surely contradicts there claim that precipitation patterns have changed with time. Thirdly, Figure 6 shows little agreement in the trends during 1981-2007 for the four datasets, with large differences in magnitude as well as even differences in sign.

Bollasina et al., Anthropogenic aerosols and the weakening of the South Asian Summer Monsoon, Science, 2011.

Major comment 2: The description of the datasets is poor and overly generalised, and does not focus enough on the study regions. For example, the description of the IMD dataset does not say explicitly how many stations are used in the study region, and what altitudes. Instead, vague language such as 'less stations near the borders of India and the in the northern part' are used. This is insufficient information to make any robust judgement of the veracity of the data. This type of vague description is continued for APHRODITE. For example, in the description of APHRODITE, evidence of its representativeness of precipitation distribution is given by the claim that it is better than the MRI/JMA AGMC model – when in fact it is data that should be used to ground-truth models, and not the other way around. My understanding of both these datasets is that due to the sparcity of gauge measurements in the Himalayas, and particularly the lack of measurements at high altitudes, that these datasets are highly biased. In the description of ERA-Interim, it is stated that 'the spatial resolution of the ERA-interim dataset is limited in representing the spatial variability'. If this is not representative of precipitation, then why is it being used? Moreover, a vague statement that 'precipitation is adjusted based on GPCP v2.1. before release' is included. What is GPCP data? How does this affect the representation of precipitation over the Himalayas in

ERA-interim? None of these questions are answered. Finally, it was odd that the WRF model run used the configuration of EURO-CORDEX, rather than that recommended by e.g. papers by Maussion et al. (2011) or Collier and Immerzeel (2015) which focused on the Himalayas. This unfortunately gives the impression that the authors were using the model has a 'black box', and had little understanding of regional atmospheric modelling. This is reinforced by statements such as 'The ERA-Interim and WRF datasets are products with different dynamical models' and referring to both of these as 'the products from dynamic models do not suffer from an undercatch', which suggests that the authors aren't properly aware of the considerable differences between reanalysis products and numerical weather prediction modelling. The claim that the model run was not optimised was 'due to the complex orography' is unfounded, as studies such as Maussion et al., Collier et al. have shown that the choice of model grids and physics parameterisations is critical. There are also no details as to the spatial resolution of the WRF model, and claims such as 'the climate model has been proved to produce the regional precipitation at a fine scale' are for models running at kilometre scale for small regions around 100 km in size (See Collier and Immerzeel).

Collier and Immerzeel, High resolution modelling of atmospheric dynamics in the Nepalese Himalayas, JGR, 2015.

Major comment 3: The Abstract begins by saying that 'data scarcity is the biggest problem ... in the Himalayas' and that 'high quality precipitation data are difficult to obtain'. Yet the paper never properly addresses which of the four datasets is, despite their deficiencies, best able to represent Himalayan precipitation patterns. This might have been a worthwhile objective. Indeed, the abstract states that 'all the datasets can give a good overview of the precipitation'. How can this be possible when one of your datasets is ERA-Interim and another is WRF-based downscaling of ERA-Interim? It is unclear how this conclusion is reached, other than the broad generalisation that all the datasets show a wetter summer compared to winter. Moreover, does this mean that all datasets would give broadly the same answers if they were used as input to hydrology models? Additionally, many of the findings are well known, such as 'the highest precipitation locates at the foothill of the mountains and stretches from southeast to northwest'. Some of the results seemed distinctly unoriginal. The authors cite the Bookhagen and Burbank (2006) study, which did a very thorough job of describing precipitation characteristics in the Himalayas. I was unsure whether one of your aims was to show which datasets could recreate their findings? Also, any results which claimed to show something original, such as changes in precipitation, were highly flawed (see comment above).

Major comment 4: The manuscript is poorly organised, and lacking depth and understanding of the topic. For example, much of the results section is filled with material which should have been in either sections 1 or 2. The authors cite the study of Li et al. (2016) as indirectly proving that the WRF model can realistically simulate precipitation as it was able to force a hydrological model which was able to simulate discharge values. However, possibly the hydrology model was tuned to get this result? Moreover, the WRF model is highly sensitive to choice of physics and model setup (as the study by Maussion et al. (2011) shows), so much more details should have provided of how your model setup agrees with that of Maussion et al. This again illustrates that the authors are not suitably experienced in modelling to have included the WRF output.

---

## Referee Comment (RC2) · Anonymous Referee #2 · 5 Jul 2017

Precipitation data are a key input in hydrologic modeling and the present paper compares four precipitation datasets which include data obtained by ground based measurements, interpolation, and reanalysis data. The paper addresses an important issue faced by hydrologists. I have the following comments on the paper. 1. The strengths and weaknesses of the four type of gridded precipitation datasets explained in lines 15 to 30 on page 2 can be better explained by means of a table. Each row of this table may correspond to a particular dataset and the columns could the how the data is obtained, its strengths, and weaknesses. 2. In this paper, IMD dataset at 1 degree grid has been used. Currently, data at 0.25 degree resolution are also available. 3. There is a view that the precipitation data obtained from instrumented stations does not reveal the actual values over a catchment in Western Himalaya because the network of stations does not have the desired density and most stations are located in valleys. Thus, the actual precipitation in the hill tops is not known. 4. Authors mention that the four datasets are similar in terms of spatial and temporal variation but there is very large variation in absolute values from 497 to 819 mm/year. Given this, a reader would expect clear view from the authors: a) what is their assessment of mean annual rainfall, and b) which dataset(s) can be used in applications such as water yield assessment, flood forecasting, climate change impact assessment, and so on.
* * *

---

## Referee Comment (RC3) · Anonymous Referee #3 · 6 Jul 2017

This paper is dealing with an interesting data challenge in Himalayan region through comparison of four globally available gridded precipitation data sets. Though this work is interesting and publishable in terms of regional importance of Himalaya, illustrations in present format are not very strong. Major demerits are 1. This paper looks like a quick dissemination with limited analysis from authors (some figures just direct illustration of latitude wise and seasonal raw data). 2. Lack of appropriate inter-comparison technique which ensures comparability of spatial patterns with different grid size. 3. Lack of detailed discussions on spatial patterns and issues of scale. 4. Lack of verification indicators (example: POD, TS, FAR, FBI etc.) in comparison with available rain gauge data or IMD data as reference.

Some specific comments are given below:

[Figure]

As this work is dealing with comparison gridded data sets with varying spatial resolution and other features, the discussion needs to be strengthened incorporating these uncertainty aspects and how reliable/meaningful these absolute precipitation values are, which are used for comparison.

Need some consistency in acronyms used in this text ( e.g.: authors have used both AGMC and AGCM)

Bit more clarity is needed in the description of seasons in the study region (e.g.: in Page 3 authors have considered November-April period as winter, and May to October as summer).

More clarity and justifications through references are required to strengthen "do not suffer from the undercatch problem" explained in the page number 3.

Better clarity is needed on the selection of 5 year time slices (1981-1985, 2003-2007) for comparison. Why seasonal comparison is limited to selected two months (Page 6) only?

Page 2 Paragraph 2. This part of text emphases more on to a specific project and associated difficulties- it appears to limit the scope of this work. It would be more appropriate if you define these site-specific difficulties and rationale of this paper through proper references from Himalayan region than describing it as a 'INDICE project' related issue.

I am doubtful about the usefulness of west-to-east latitude based precipitation comparisons for hydrologists in the region. It would be more useful for the hydrologic research community if you could include precipitation PDFs comparisons of larger river basins (Indus River and the upper Ganges River) in this study domain

Titles of Figure 7 and 8: you need to clearly write the details of data points with different colors.

---

## Author Comment (AC1) · 4 Oct 2017

**Reply to reviewers' comments**

Dear Editor and Reviewers:

We thank the handling Editor and the reviewers for your comments and suggestions concerning our manuscript entitled "Precipitation Pattern in the Western Himalayas revealed by Four Datasets" (Manuscript Number: hess-2017-296). These comments are all valuable and very helpful not only for improving this paper but also beneficial for our research in general. We have carefully studied these comments and will address them in making revisions. The point-by-point responses to each of the comments are presented as follows.

**Anonymous Referee #1**

Major comment 1: The methodology of attempting to distinguish precipitation trends from the four types of dataset is not scientifically valid. Firstly, there is no attempt to calculate the inter-annual variability of precipitation for each of the datasets. Therefore, it is impossible to tell whether the changes between the 2003-2007 and 1981-1985 periods are meaningful. There was also no justification for why these periods were even chosen. It is also impossible to tell whether choosing five years for each period is large enough to capture the precipitation representative of the 1980's and 2000's, i.e. these periods could easily have been anomalous. There is also no physical justification given as to whether the e.g. increase in summer precipitation during these two periods is physically consistent with either dynamic or thermodynamic large-scale changes, such as in response to the observed weakening of summer monsoonal precipitation (e.g. Bollasina et al. 2011). The differences also aren't quantified, and are referred to in the Abstract as 'an increase in summer and a decrease in winter with large variations'. The fact that there are large differences between the datasets is also a cause for concern, with the differences between datasets being possibly greater than the magnitude of the differences between the two periods.

Reply: Thanks for the reviewer's comments. There are three questions and we answer each individually.

For first question, we will follow the reviewer's advice, we will calculate and compare the inter-annual variability of precipitation for each of the datasets, a discussion of which will be included in the revision.

For second question, we agree that it is not clear why the 2003-2007 and 1981-1985 periods are chosen, and whether they are representative. As we stated in the manuscript that there are large differences among the datasets with respect to seasonality and spatial pattern. Such differences have implications for hydrology and glaciers. In the revised version we will follow reviewer's advice to provide more in-depth discussion on this aspect. We will make more comprehensive analysis (annual and seasonal variability), and extend the analysis to the whole study period with moving time-slices. Such analysis will give robust results and uncertainty existing in trend analysis.

For third question, we will look at some other variables, such as wind speed and air pressure in the ERA-interim to find possible causes. We will do a more thorough literature study on the changes in the Western Himalayan Region and to find if the changes will continue in the future.

Secondly, the authors showed that the linear precipitation trend was insignificant, which surely contradicts their claim that precipitation patterns have changed with time.

Reply: Sorry that we failed to state that clear enough in the original version. Linear trend for yearly total precipitation is not significant, but precipitation patterns as reflected by seasonality and spatial variability have changed more. We will provide more explanation on this aspect in the revised version.

Thirdly, Figure 6 shows little agreement in the trends during 1981-2007 for the four datasets, with large differences in magnitude as well as even differences in sign.

Reply: This is true. As we mentioned in the reply of previous comment, there are large differences among the datasets with respect to seasonality and spatial pattern. Such differences have implications for hydrology and glaciers. We will put more effort in the revised version to provide more in-depth discussion on this aspect. Additionally, we will do trend analysis at seasonal scales for more robust results. The overlap period of the four datasets is 27 years. However, the ERA-interim and the IMD datasets cover respectively the periods of 1979-2016 and 1951-2007. We will look at the long-term changes of these two datasets to see if the trends of the study period agree with the long term trends.

Bollasina et al., Anthropogenic aerosols and the weakening of the South Asian Summer Monsoon, Science, 2011.

Major comment 2: The description of the datasets is poor and overly generalised, and does not focus enough on the study regions. For example, the description of the IMD dataset does not say explicitly how many stations are used in the study region, and what altitudes. Instead, vague language such as 'less stations near the borders of India and the in the northern part' are used. This is insufficient information to make any robust judgement of the veracity of the data. This type of vague description is continued for APHRODITE. For example, in the description of APHRODITE, evidence of its representativeness of precipitation distribution is given by the claim that it is better than the MRI/JMA AGMC model – when in fact it is data that should be used to ground-truth models, and not the other way around. My understanding of both these datasets is that due to the sparcity of gauge measurements in the Himalayas, and particularly the lack of measurements at high altitudes, that these datasets are highly biased. In the description of ERA-Interim, it is stated that 'the spatial resolution of the ERA-interim dataset is limited in representing the spatial variability'. If this is not representative of precipitation, then why is it being used? Moreover, a vague statement that 'precipitation is adjusted based on GPCP v2.1. before release' is included. What is GPCP data? How does this affect the representation of precipitation over the Himalayas in ERA-interim? None of these questions are answered.

Reply: Thanks for the professional and constructive comments. We agree and we apologize for the vague description and we will add more details in the revised version on the description of all the datasets used in the study.

Finally, it was odd that the WRF model run used the configuration of EURO-CORDEX, rather than that recommended by e.g. papers by Maussion et al. (2011) or Collier and Immerzeel (2015) which focused on the Himalayas. This unfortunately gives the impression that the authors were using the model has a 'black box', and had little understanding of regional atmospheric modelling. This is reinforced by statements such as 'The ERA-Interim and WRF datasets are products with different dynamical models' and referring to both of these as 'the products from dynamic models do not suffer from an undercatch', which suggests that the authors aren't properly aware of the considerable differences between reanalysis products and numerical weather prediction modelling. The claim that the model run was not optimised was 'due to the complex orography' is unfounded, as studies such as Maussion et al., Collier et al. have shown that the choice of model grids and physics parameterisations is critical. There are also no details as to the spatial resolution of the WRF model, and claims such as 'the climate model has been proved to produce the regional precipitation at a fine scale' are for models running at kilometre scale for small regions around 100 km in size (See Collier and Immerzeel).

Collier and Immerzeel, High resolution modelling of atmospheric dynamics in the Nepalese Himalayas, JGR, 2015.

Reply: Thanks for the comment, and sorry that we did not state it clear enough. In the revised version we will provide more discussion concerning the model configuration and also explain the difference with the nice work of Maussion et al. (2011). The WRF model runs at $0.15° \times 0.15°$ grid. Maussion et al. (2011) indeed did a very good comparison of model configuration at the Tibet Planet (TiP) area, which is around 10 degree eastern of our study area. However, they conclude, "Our study reveals that there is nothing like an optimal model strategy applicable for the high-altitude TiP, its fringing high-mountain areas of extremely complex topography and the low-altitude land and sea regions from which much of the precipitation on the TiP is originating.  The choice of the physical parameterization scheme will thus be always a compromise depending on the specific purpose of a model simulation. Our study demonstrates the high importance of orographic precipitation, but the problem of the orographic bias remains unsolved since reliable observational data are still missing". Our WRF configuration has also been used in other projects, which focus on the western Asia and our results show that this configuration is reliable at the study area. We use the same microphysics and land surface as their reference experiments and Li et al. (2017) has tested the microphysics, cumulus and land surface scheme. This part of the text will be largely rewritten in accordance with reviewer's comment and advice.

Li, L., Gochis, D. J., Sobolowski, S., & Mesquita, M. D. S. (2017). Evaluating the present annual water budget of a Himalayan headwater river basin using a high-resolution atmosphere-hydrology model. *Journal of Geophysical Research: Atmospheres*, *122*(9), 4786–4807. article. https://doi.org/10.1002/2016JD026279

Major comment 3: The Abstract begins by saying that 'data scarcity is the biggest problem . . . in the Himalayas' and that 'high quality precipitation data are difficult to obtain'. Yet the paper never properly addresses which of the four datasets is, despite their deficiencies, best able to represent Himalayan precipitation patterns. This might have been a worthwhile objective. Indeed, the abstract states that 'all the datasets can give a good overview of the precipitation'. How can this be possible when one of your datasets is ERA-Interim and another is WRF-based downscaling of ERA-Interim? It is unclear how this conclusion is reached, other than the broad generalisation that all the datasets show a wetter summer compared to winter. Moreover, does this mean that all datasets would give broadly the same answers if they were used as input to hydrology models? Additionally, many of the findings are well known, such as 'the highest precipitation locates at the foothill of the mountains and stretches from southeast to northwest'. Some of the results seemed distinctly unoriginal. The authors cite the Bookhagen and Burbank (2006) study, which did a very thorough job of describing precipitation characteristics in the Himalayas. I was unsure whether one of your aims was to show which datasets could recreate their findings? Also, any results which claimed to show something original, such as changes in precipitation, were highly flawed (see comment above).

Reply: We agree with reviewer's concern, and we apologize that we failed to address our important objectives clear enough, i.e. to look at differences among datasets and their implications on hydrology and glaciers. To achieve this, we will add a new section, in which we will select available catchments in The Global Runoff Data Centre (GRDC, http://www.bafg.de/GRDC/EN/01_GRDC/grdc_node.html;jsessionid=0F83978153B3C0221 4DBA224F2084914.live21302) and the Integrated Hydrological Data Book (CWC, http://www.cwc.nic.in/ISO_DATA_Bank/ISO_Home_Page.htm) within the study area. We will analyze runoff data and actual evaporation from MODIS. We will answer the question like which datasets are more reliable in water balance assessment and how much is uncertainty.

Major comment 4: The manuscript is poorly organised, and lacking depth and understanding of the topic. For example, much of the results section is filled with material which should have been in either sections 1 or 2. The authors cite the study of Li et al. (2016) as indirectly proving that the WRF model can realistically simulate precipitation as it was able to force a hydrological model which was able to simulate discharge values. However, possibly the hydrology model was tuned to get this result? Moreover, the WRF model is highly sensitive to choice of physics and model setup (as the study by Maussion et al. (2011) shows), so much more details should have provided of how your model setup agrees with that of Maussion et al. This again illustrates that the authors are not suitably experienced in modelling to have included the WRF output.

Reply: We will reorganize the structure of the paper, into Introduction, Study area and data, Methodology, Results and Discussion, Conclusions. In the revised version, the Results will be divided into two sub-sections. One is on precipitation data, which consists of the current sections 1 and 2. Another is on implications about impacts, which consists of the current section 3 and a new section, which is usage of runoff and actual evaporation of selected

catchments. The WRF-hydro is physically based land-surface process model. The tuning is used to find correct values for parameters, which is due to lack of measurements of vegetation, soil and river channel characteristics. In the revised version, we will give more details of the WRF model setting and will provide the configuration file as support material.

**Anonymous Referee #2**

Precipitation data are a key input in hydrologic modeling and the present paper compares four precipitation datasets which include data obtained by ground based measurements, interpolation, and reanalysis data. The paper addresses an important issue faced by hydrologists. I have the following comments on the paper.

Reply: Thanks for reviewer's positive evaluation in general, and the specific comments which are detailed below.

1. The strengths and weaknesses of the four type of gridded precipitation datasets explained in lines 15 to 30 on page 2 can be better explained by means of a table. Each row of this table may correspond to a particular dataset and the columns could the how the data is obtained, its strengths, and weaknesses.

Reply: We thank the reviewer for this good suggestion and we will add a table following the advice of the reviewer.

2. In this paper, IMD dataset at 1 degree grid has been used. Currently, data at 0.25 degree resolution are also available.

Reply: This research is based on 1-degree grid data and we obtained this dataset from our Indian research partner. This resolution is comparable with other datasets used in the study. Thanks for introducing the 0.25° IMD dataset, which is not available for the authors at this moment and will be considered in the future study.

3. There is a view that the precipitation data obtained from instrumented stations does not reveal the actual values over a catchment in Western Himalaya because the network of stations does not have the desired density and most stations are located in valleys. Thus, the actual precipitation in the hill tops is not known.

Reply: The actual precipitation is largely unknown at the hill tops. Some methods are available to partly overcome this problem. One is to interpolate the ground observations with elevation and undercatch correction. Numerical models are also valuable to estimate precipitation at high mountains, especially where observations are not reliable or not available. This manuscript is an attempt to compare four datasets produced by different methods.

4. Authors mention that the four datasets are similar in terms of spatial and temporal variation but there is very large variation in absolute values from 497 to 819 mm/year. Given this, a reader would expect clear view from the authors: a) what is their assessment of mean annual

rainfall, and b) which dataset(s) can be used in applications such as water yield assessment, flood forecasting, climate change impact assessment, and so on.

Reply: We agree these questions are very important and we must give a clear view. To achieve this, we will add a new section, in which we will select available catchments in The Global Runoff Data Centre (GRDC, http://www.bafg.de/GRDC/EN/01_GRDC/grdc_node.html;jsessionid=0F83978153B3C0221 4DBA224F2084914.live21302) and the Integrated Hydrological Data Book (CWC, http://www.cwc.nic.in/ISO_DATA_Bank/ISO_Home_Page.htm) within the study area. We will analyze runoff data and actual evaporation from MODIS. We will try to answer the question like which datasets are more reliable in water balance assessment and how much is uncertainty.

**Anonymous Referee #3**

This paper is dealing with an interesting data challenge in Himalayan region through comparison of four globally available gridded precipitation data sets. Though this work is interesting and publishable in terms of regional importance of Himalaya, illustrations in present format are not very strong.

Reply: Thanks for reviewer's positive evaluation in general, and we appreciate very much the constructive comments which are detailed below.

Major demerits are 1. This paper looks like a quick dissemination with limited analysis from authors (some figures just direct illustration of latitude wise and seasonal raw data).

Reply: We will add more details in response to specific comments and a new section about uncertainty based on runoff and evaporation. More details of our response are given below.

2. Lack of appropriate inter-comparison technique which ensures comparability of spatial patterns with different grid size.

Reply: The precipitation datasets are at different grid size and it is quite challenging. The results show in forms of spatial maps, individual grids and regional mean. We will additionally use histogram as well as different quantiles to discuss in more details of the results.

3. Lack of detailed discussions on spatial patterns and issues of scale.

Reply: Agree. We will add more details about the spatial pattern. More details about issues of scale will be added in the section of spatial pattern as well as in a new section about uncertainty based on runoff and evaporation (MODIS) data for available catchments in The Global Runoff Data Centre (GRDC, http://www.bafg.de/GRDC/EN/01_GRDC/grdc_node.html;jsessionid=0F83978153B3C0221 4DBA224F2084914.live21302) and the Integrated Hydrological Data Book (CWC, http://www.cwc.nic.in/ISO_DATA_Bank/ISO_Home_Page.htm).

4. Lack of verification indicators (example: POD, TS, FAR, FBI etc.) in comparison with available rain gauge data or IMD data as reference.

Reply: Thanks for the comment and suggestion. Some of the verification indicators as used in Sikder & *Hossain (2016)*, i.e., POD: the probability of detection. TS: threat Score. FAR: the false alarm ratio. FBI: the frequency bias index, will be used in the revised version. These indexes were used to assess forecasting results with observations. Here we do not have "true observations"- rain gauge data. The IMD dataset will be used as reference although it is interpolated by the India Meteorological Department from rain gauge data. During the period of 1981-2007, the average number of stations per grid point (1×1) varies from 0.2 to 4.4. Fewer stations near the borders of India and in the northern part of the study area and observations are available near the latitude of 35.5 N and its north.

Sikder, S., & Hossain, F. (2016). Assessment of the weather research and forecasting model generalized parameterization schemes for advancement of precipitation forecasting in monsoon-driven river basins. *Journal of Advances in Modeling Earth Systems*, *8*(3), 1210–1228. article. http://doi.org/10.1002/2016MS000678

Some specific comments are given below:

As this work is dealing with comparison gridded data sets with varying spatial resolution and other features, the discussion needs to be strengthened incorporating these uncertainty aspects and how reliable/meaningful these absolute precipitation values are, which are used for comparison.

Reply: Thanks for the comment. To address this issue, in the revised version we will add a new section, in which, we will select available catchments in The Global Runoff Data Centre (GRDC, http://www.bafg.de/GRDC/EN/01_GRDC/grdc_node.html;jsessionid=0F83978153B3C02214DBA224F2084914.live21302) and the Integrated Hydrological Data Book (CWC, http://www.cwc.nic.in/ISO_DATA_Bank/ISO_Home_Page.htm) within the study area. We will analyze runoff data and actual evaporation from MODIS. We will answer the question like which datasets are more reliable and how much is uncertainty.

Need some consistency in acronyms used in this text ( e.g.: authors have used both AGMC and AGCM)

Reply: This is a typos. We will revise it in the revised version.

Bit more clarity is needed in the description of seasons in the study region (e.g.: in Page 3 authors have considered November-April period as winter, and May to October as summer).

Reply: We apologize for this ambiguity. Here it refers to summer monsoon season and winter monsoon season. The analysis is based on northern meteorological seasons (spring: March to May; summer: June to August; Autumn: September to November; Winter: December to February). We will revise it in the revised version.

More clarity and justifications through references are required to strengthen "do not suffer from the undercatch problem" explained in the page number 3.

Reply: Thanks. In the revised version we will clarify it. Undercatch means the amount of precipitation is not measured by gauges mostly caused by wind turbulence especially for snow. The undercatch has a significant effect for unshielded and single gauges at high latitude and high altitude. The undercatch applies to most types of precipitation observations and their products. Here we meant that Regional climate models simulate precipitation based on mathematical equations and the results are considered not affected by undercatch problem.

Better clarity is needed on the selection of 5 year time slices (1981-1985, 2003-2007) for comparison.

Reply: We will clarify that the trend is analyzed by the Mann-Kendall test. It shows winter became drier and summer became wetter during the period of 1981 to 2007. However, few (May by the WRF dataset; June by the IMD and ERA-interim datasets) of them are statistically significant at the 95% confidence level (Figure 6 and Page 7). We believe the trends will have negative impact on mass balance of glaciers and comparison between the first and the last five years will give a quantitative picture of the impact. But in the revised version we will use the whole period to quantify the changes.

Why seasonal comparison is limited to selected two months (Page 6) only?

Reply: The seasonal comparison is shown in Figure 3 whereas these months are purposefully selected to have a better visual impression of spatial distribution and interaction with topography. JA (July and August) and ND (November and December) have respectively highest and lowest precipitation and these months clearly show the changes as illustrated in Figure 5.

Page 2 Paragraph 2. This part of text emphases more on to a specific project and associated difficulties- it appears to limit the scope of this work. It would be more appropriate if you define these site-specific difficulties and rationale of this paper through proper references from Himalayan region than describing it as a 'INDICE project' related issue.

Reply: We agree with the reviewer and will revise the manuscript accordingly.

I am doubtful about the usefulness of west-to-east latitude based precipitation comparisons for hydrologists in the region. It would be more useful for the hydrologic research community if you could include precipitation PDFs comparisons of larger river basins (Indus River and the upper Ganges River) in this study domain

Reply: This figure aims to show interactions between atmosphere and topography in JA (July and August, high precipitation months) and ND (November and December, low precipitation months). As shown in the figure, the precipitation changes at different locations and the changes vary among the four datasets. We agree with the reviewer that precipitation PDFs for

catchments are very useful for hydrologists and we will add such analysis in the revised version.

Titles of Figure 7 and 8: you need to clearly write the details of data points with different colors.

Reply: In Figure 7, the blue color shows the period 1981-1985 and the orange shows the period 2003-2007. These are shown as in the legend. In Figure 8, the colors differentiate between below or above zero. We will add more details in the revised version.

---

## Author Response (AR1)

**Editor Decision: Publish subject to revisions (further review by editor and referees)** (01 Nov 2017) by Ian Holman
Comments to the Author:
All reviewers recognise that the scope of this manuscript represents an interesting and relevant topic, relating to the uncertainty in precipitation data products in a region of relatively poor instrumental data. However, all reviewers identify important inadequacies in the current version, which can be summarised as a :
• Lack of in-depth spatio-temporal comparison and analysis of the absolute precipitation values within the datasets;

Reply: We first thank the editor and the reviewers for your positive evaluation of our work in general, and we appreciate very much the professional and constructive comments raised by reviewers and editor.

In response to Editor's first comment of providing in-depth spatio-temporal comparison and analysis of absolute precipitation values within the datasets, we further did trend analysis by the Theil-Sen median method and significance test by the non-parametric Mann-Kendall method for areal mean of each month and every individual grid for summer and winter. The results are summarized in bar-plots, density plots as well as spatial maps. We also analyzed precipitation across latitude and elevations. We further added precipitation-evaporation-runoff analysis in the Beas catchment by three nested sub-catchments. Based on the in-depth comparison and analysis, finally, we recommended the APHRODITE data and the WRF data for the study regions due to matched scales with hydrological processes and considerable spatial variation.

• Lack of rationale for the selection and length (5 years) of the inter-comparison period, and whether any differences found are meaningful in light of inter-annual variability and large-scale meteorological forcings;

Reply: We extended trend analysis by the robust Theil-Sen median method and significance test by the non-parametric Mann-kendall test, to study the inter-annual variability of the whole period. The five-years' comparison was used for illustrative purpose to show temperature changes with respect to elevation for glacier survival analysis. Because temperature products are more reliable and less uncertain than precipitation products. It is also well known that temperature in the Great Himalayas Region increases fast since 1980s. There is no need to go through many temperature products and to do trend analysis and significance test. Three trends of the months were significant at the 5% confidence level and and overall 10% of grids were significant at the 10% confidence level.

It is difficult to conclude why the Northern India of the Western Himalayas showed an increase in summer precipitation. However, Bollasina et al. (2011) found the same increasing monsoon precipitation in the Northern India, whereas decreasing in the Central Asia. They used a series of climate model experiments,  concluded that such pattern was a robust outcome of a slowdown of the tropical meridional overturning circulation, which could be attributed mainly to  human-influenced aerosol emissions. Therefore, we believe that the trends will continue and become more significant with time if greenhouse gas emission continues as usual.

• Lack of consideration of the implications of the different spatial scales (grid sizes) of the data products

Reply: Thanks for the good suggestion and challenging comment. Here we used two spatial scales, i.e. aggregated areal mean and individual grid. The results were shown in scatter-plot, bar-plot, density plot as well as spatial maps. Here we selected three nested catchments in the Beas catchment with respect of different elevation and climate and analysed the precipitation-evaporation-runoff relationship, which was at the scale of hydrological process. Based on the in-depth comparison and analysis, for hydrological

studies, we recommend the APHRODITE data and the WRF data due to matched scales with hydrological processes and considerable spatial variation.

• Lack of any benchmarking to determine which is the 'best' dataset for use in hydrological modelling and other types of studies, particularly given the uncertainty in deriving actual precipitation amounts from instruments that do not adequately capture spatial (and vertical) heterogeneity in these data sparse regions

Reply: Thanks for the comment, which points out the challenging nature of study in spatial (and vertical) heterogeneity and data sparse regions. We selected three nested catchments in the Beas catchment with respect to different elevation and climate. We indeed tried The Global Runoff Data Centre for more catchments, but the overlap period of runoff and precipitation was too short. The comparison showed precipitation was underestimated. Though the relationships were different, all datasets were consistent in terms of temporal changes and errors were systematic within each dataset. We also analyzed the MODIS Global Evapotranspiration Project annual mean actual evaporation, but the values were far from truth. For hydrological studies, we recommend the APHRODITE data and the WRF data due matched scales with hydrological processes and considerable spatial variation.

The authors' response to the reviewers lacks specificity but suggests that they will address the concerns raised - for the paper to be published, the identified weaknesses will need to be comprehensively addressed. They identify that they do not have access to the IMD 0.25o dataset for use as a benchmark, but propose to utilise observed river flow and MODIS ET data. This is an interesting approach, although its utility will depend upon the spatial scale of the river basins with available data (i.e. whether their extent leads to excessive spatial averaging) and the importance of agricultural irrigation in the river basins (and therefore how different gauged flows are from naturalised flows)

Non-public comments to the Author:
This paper has the potential to be a very useful and widely cited paper given the uncertainties in precipitation data in this region, if the authors can adequately address the weaknesses identified by the papers.

Reply: We are very much grateful to the Editor's affirmation and encouragement, we believe we have done our best in revising the paper following reviewers' and editor's comments and advice, given the limited availability of data in the region. Limitation of data, on other hand, strengthened the necessity and usefulness of such a study, as the Editor pointed out.

**Reply to reviewers' comments**

Dear Editor and Reviewers:

We thank the handling Editor and the reviewers for your comments and suggestions concerning our manuscript entitled "Precipitation Pattern in the Western Himalayas revealed by Four Datasets" (Manuscript Number: hess-2017-296). These comments are all valuable and very helpful not only for improving this paper but also beneficial for our research in general. We have carefully studied these comments and will address them in making revisions. The point-by-point responses to each of the comments are presented as follows.

**Anonymous Referee #1**

Major comment 1: The methodology of attempting to distinguish precipitation trends from the four types of dataset is not scientifically valid. Firstly, there is no attempt to calculate the inter-annual variability of precipitation for each of the datasets. Therefore, it is impossible to tell whether the changes between the 2003-2007 and 1981-1985 periods are meaningful. There was also no justification for why these periods were even chosen. It is also impossible to tell whether choosing five years for each period is large enough to capture the precipitation representative of the 1980's and 2000's, i.e. these periods could easily have been anomalous. There is also no physical justification given as to whether the e.g. increase in summer precipitation during these two periods is physically consistent with either dynamic or thermodynamic large-scale changes, such as in response to the observed weakening of summer monsoonal precipitation (e.g. Bollasina et al. 2011). The differences also aren't quantified, and are referred to in the Abstract as 'an increase in summer and a decrease in winter with large variations'. The fact that there are large differences between the datasets is also a cause for concern, with the differences between datasets being possibly greater than the magnitude of the differences between the two periods.

Reply: Thanks for the reviewer's comments. There are three questions and we answer each individually.

For first question, we calculated and compared the inter-annual variability of precipitation for each of the datasets by the Theil-Sen median method and the non-parametric Mann-Kendall method for every individual grid as shown in Section 3.2 Temporal variations and changes. The results showed an overall increase in summer precipitation and decrease in winter precipitation.

For second question, we agree that it is not clear why the 2003-2007 and 1981-1985 periods were chosen, and whether they were representative. Therefore, we analyzed the whole period. This was done for areal mean of all months and every individual grid for summer and winter. The two periods, first and last five-years were only used for illustrative purpose to show temperature changes with respect to elevation for glacier survival, because temperature measurements were of higher quality and less uncertain than precipitation. There was no need to go though many datasets. It is also well know that temperature is increasing fast in the Great Himalayas Region, so there was no need for the trend analysis and the significant test.

It is difficult to conclude why the Northern India of the Western Himalayas showed an increase in summer precipitation. However, Bollasina et al. (2011) found the same increasing monsoon precipitation in the Northern India, whereas decreasing in the Central Asia. They used a series of climate model experiments, concluded that such pattern was a robust outcome of a slowdown of the tropical meridional overturning circulation, which could be attributed mainly to human-influenced aerosol emissions. This was also in the manuscript.

Secondly, the authors showed that the linear precipitation trend was insignificant, which surely contradicts their claim that precipitation patterns have changed with time.

Reply: Sorry that we failed to state that clear enough in the original version. The trend analysis used the Theil-Sen median method and the significance test used the non-parametric Mann-Kendall test. Both test were done for areal mean for each months and every individual grid for summer and winter. Most of the trends, were not significant at the 5% confidence level by the Mann-Kendall test, but some are indeed significant, i.e. three trends for monthly areal mean at the 5% confidence level and overall 10% trends for grids at the 10% confidence level. As we know, statistic test results are affected by sample size, and we believe the trends will become more significant with time if greenhouse gas emission continues as usual. We have added some more discussion on this aspect.

Thirdly, Figure 6 shows little agreement in the trends during 1981-2007 for the four datasets, with large differences in magnitude as well as even differences in sign.

Reply: This is true. As we mentioned in the reply of previous comment, there are large differences among the datasets with respect to absolute magnitude. However, their spatial pattern, intra-annual variability as well as relationships with runoff are similar. All datasets show similar trend of winter precipitation in the northern part as shown in Figure 10. To find difference and similarity among the datasets and discuss their implication in hydrology are the objective of this manuscript.

Bollasina et al., Anthropogenic aerosols and the weakening of the South Asian Summer Monsoon, Science, 2011.

Major comment 2: The description of the datasets is poor and overly generalised, and does not focus enough on the study regions. For example, the description of the IMD dataset does not say explicitly how many stations are used in the study region, and what altitudes. Instead, vague language such as 'less stations near the borders of India and the in the northern part' are used. This is insufficient information to make any robust judgement of the veracity of the data. This type of vague description is continued for APHRODITE. For example, in the description of APHRODITE, evidence of its representativeness of precipitation distribution is given by the claim that it is better than the MRI/JMA AGMC model – when in fact it is data that should be used to ground-truth models, and not the other way around. My understanding of both these datasets is that due to the sparcity of gauge measurements in the Himalayas, and particularly the lack of measurements at high altitudes, that these datasets are highly biased. In the description of ERA-Interim, it is stated that 'the spatial resolution of the ERA-interim dataset is limited in representing the spatial variability'. If this is not representative of precipitation, then why is it being used? Moreover, a vague statement that 'precipitation is adjusted based on GPCP v2.1. before release' is included. What is GPCP data? How does this affect the representation of precipitation over the Himalayas in ERA-interim? None of these questions are answered.

Reply: Thanks for the professional and constructive comments. We apologize for the vague description and have added more details in the revised version on the description of all the datasets used in the study as well as about GPCP. This part of the manuscript has been largely improved.

Finally, it was odd that the WRF model run used the configuration of EURO-CORDEX, rather than that recommended by e.g. papers by Maussion et al. (2011) or Collier and Immerzeel (2015) which focused on the Himalayas. This unfortunately gives the impression that the authors were using the model has a 'black box', and had little understanding of regional atmospheric modelling. This is reinforced by statements such as 'The ERA-Interim and WRF datasets are products with different dynamical models' and referring to both of these as 'the products from dynamic models do not suffer from an undercatch', which suggests that the authors aren't properly aware of the considerable differences between reanalysis products and numerical weather prediction modelling. The claim that the model run was not optimised was 'due to the complex orography' is unfounded, as studies such as Maussion et al., Collier et al. have shown that the choice of model grids and physics parameterisations is critical. There are also no details as to the spatial

resolution of the WRF model, and claims such as 'the climate model has been proved to produce the regional precipitation at a fine scale' are for models running at kilometre scale for small regions around 100 km in size (See Collier and Immerzeel).

Collier and Immerzeel, High resolution modelling of atmospheric dynamics in the Nepalese Himalayas, JGR, 2015.

Reply: Thanks for the comment, and sorry that we did not state it clear enough. In the revised version, we provided more discussion and information about the WRF model configuration in manuscript and uploaded the completely model-setting file as supplementary file. Maussion et al. (2011) indeed did a very good comparison of model configuration at the Tibet Planet (TiP) area, which is around 10 degree eastern of our study area. However, they conclude, "Our study reveals that there is nothing like an optimal model strategy applicable for the high-altitude TiP, its fringing high-mountain areas of extremely complex topography and the low-altitude land and sea regions from which much of the precipitation on the TiP is originating. The choice of the physical parameterization scheme will thus be always a compromise depending on the specific purpose of a model simulation. Our study demonstrates the high importance of orographic precipitation, but the problem of the orographic bias remains unsolved since reliable observational data are still missing". Our WRF configuration has also been used in other projects, which focus on the western Asia and our results show that this configuration is reliable at the study area. We used the same microphysics and land surface as their reference experiments and Li et al. (2017) has tested the microphysics, cumulus and land surface scheme.

Li, L., Gochis, D. J., Sobolowski, S., & Mesquita, M. D. S. (2017). Evaluating the present annual water budget of a Himalayan headwater river basin using a high-resolution atmosphere-hydrology model. *Journal of Geophysical Research: Atmospheres*, *122*(9), 4786–4807. article. https://doi.org/10.1002/2016JD026279

Major comment 3: The Abstract begins by saying that 'data scarcity is the biggest problem . . . in the Himalayas' and that 'high quality precipitation data are difficult to obtain'. Yet the paper never properly addresses which of the four datasets is, despite their deficiencies, best able to represent Himalayan precipitation patterns. This might have been a worthwhile objective. Indeed, the abstract states that 'all the datasets can give a good overview of the precipitation'. How can this be possible when one of your datasets is ERA-Interim and another is WRF-based downscaling of ERA-Interim? It is unclear how this conclusion is reached, other than the broad generalisation that all the datasets show a wetter summer compared to winter. Moreover, does this mean that all datasets would give broadly the same answers if they were used as input to hydrology models? Additionally, many of the findings are well known, such as 'the highest precipitation locates at the foothill of the mountains and stretches from southeast to northwest'. Some of the results seemed distinctly unoriginal. The authors cite the Bookhagen and Burbank (2006) study, which did a very thorough job of describing precipitation characteristics in the Himalayas. I was unsure whether one of your aims was to show which datasets could recreate their findings? Also, any results which claimed to show something original, such as changes in precipitation, were highly flawed (see comment above).

Reply: We agree with reviewer's concern, and we apologize that we failed to address our important objectives clear enough, i.e. to look at differences among datasets and their implications on hydrology and glaciers. To achieve this, we added a new section "Comparison with runoff data" under "Discussions". Here we selected three nested catchments in the Beas catchment with respect of different elevation and climate. We indeed tried The Global Runoff Data Centre, but the overlap period of runoff and precipitation is too short. The comparison shows precipitation was underestimated. Though the relationships were different, all datasets were consistent in terms of temporal changes and errors were systematic within each dataset. We also analyzed the MODIS Global Evapotranspiration Project annual mean actual evaporation, but the values were far from truth.

Major comment 4: The manuscript is poorly organised, and lacking depth and understanding of the topic. For example, much of the results section is filled with material which should have been in either sections 1 or 2. The authors cite the study of Li et al. (2016) as indirectly proving that the WRF model can realistically simulate precipitation as it was able to force a hydrological model which was able to simulate discharge values. However, possibly the hydrology model was tuned to get this result? Moreover, the WRF model is highly sensitive to choice of physics and model setup (as the study by Maussion et al. (2011) shows), so much more details should have provided of how your model setup agrees with that of Maussion et al. This again illustrates that the authors are not suitably experienced in modelling to have included the WRF output.

Reply: We reorganized the structure of the paper, into Introduction, Study area and data, Methodology, Results and Discussion, Conclusions. "Results" were divided into two subsections, i.e. "Spatial variations" and "Temporal variations and changes". "Discussions" were divided into two sections, i.e. "Comparison with runoff data" and "Implications for glaciers".

The WRF-hydro is physically based land-surface process model. The tuning is used to find correct values for parameters, which is due to lack of measurements of vegetation, soil and river channel characteristics. In the revised version, we had more discussions about model setting and gave more details of the WRF model setting in Table 2. The complete settings are in a supplementary file.

**Anonymous Referee #2**

Precipitation data are a key input in hydrologic modeling and the present paper compares four precipitation datasets which include data obtained by ground based measurements, interpolation, and reanalysis data. The paper addresses an important issue faced by hydrologists. I have the following comments on the paper.

Reply: Thanks for reviewer's positive evaluation in general, and the specific comments that are detailed below.

1. The strengths and weaknesses of the four type of gridded precipitation datasets explained in lines 15 to 30 on page 2 can be better explained by means of a table. Each row of this table may correspond to a particular dataset and the columns could the how the data is obtained, its strengths, and weaknesses.

Reply: We thank the reviewer for this good suggestion and we added a table, Table 1 in the revised version.

2. In this paper, IMD dataset at 1 degree grid has been used. Currently, data at 0.25 degree resolution are also available.

Reply: This research is based on 1-degree grid data and we obtained this dataset from our Indian research partner. This resolution is comparable with other datasets used in the study. Thanks for introducing the 0.25° IMD dataset, tough it is not available for the authors at moment and will be considered in the future study.

3. There is a view that the precipitation data obtained from instrumented stations does not reveal the actual values over a catchment in Western Himalaya because the network of stations does not have the desired density and most stations are located in valleys. Thus, the actual precipitation in the hill tops is not known.

Reply: The actual precipitation is largely unknown at the hilltops. Some methods are available to partly overcome this problem. One is to interpolate the ground observations with elevation and undercatch

correction. Numerical models are also valuable to estimate precipitation at high mountains, especially where observations are not reliable or not available. This manuscript is an attempt to compare four datasets produced by different methods. We added a new table, Table 1 in the revised version to summarize their weaknesses.

4. Authors mention that the four datasets are similar in terms of spatial and temporal variation but there is very large variation in absolute values from 497 to 819 mm/year. Given this, a reader would expect clear view from the authors: a) what is their assessment of mean annual rainfall, and b) which dataset(s) can be used in applications such as water yield assessment, flood forecasting, climate change impact assessment, and so on.

Reply: We agree these questions are very important and we must give a clear view. To achieve this, we added a new section "Comparison with runoff data" under "Discussions". Here we selected three nested catchments in the Beas catchment with respect of different elevation and climate. We indeed tried The Global Runoff Data Centre, but the overlap period of runoff and precipitation is too short. The comparison shows precipitation was underestimated. Though the relationships were different, all datasets were consistent in terms of temporal changes and errors were systematic within each dataset. We also analyzed the MODIS Global Evapotranspiration Project annual mean actual evaporation, but the values were far from truth. Based on the in-depth comparison and analysis, for hydrological studies, we recommend the APHRODITE data and the WRF data due to matched scales with hydrological processes and considerable spatial variation.

**Anonymous Referee #3**

This paper is dealing with an interesting data challenge in Himalayan region through comparison of four globally available gridded precipitation data sets. Though this work is interesting and publishable in terms of regional importance of Himalaya, illustrations in present format are not very strong.

Reply: Thanks for reviewer's positive evaluation in general, and we appreciate very much the constructive comments. A point-to-point reply is below.

Major demerits are 1. This paper looks like a quick dissemination with limited analysis from authors (some figures just direct illustration of latitude wise and seasonal raw data).

Reply: We added more details in response to specific comments, new content and reorganized the structure.

2. Lack of appropriate inter-comparison technique which ensures comparability of spatial patterns with different grid size.

Reply: The precipitation datasets are at different grid size and it is quite challenging. The results show in forms of spatial maps, individual grids and regional mean as well as density plot.

3. Lack of detailed discussions on spatial patterns and issues of scale.

Reply: Agree. To achieve this, we added a new section "Comparison with runoff data" under "Discussions". Here we selected three nested catchments in the Beas catchment with respect of different elevation and climate. We indeed tried The Global Runoff Data Centre, but the overlap period of runoff and precipitation is too short. The comparison shows precipitation was underestimated. Though the relationships were different, all datasets were consistent in terms of temporal changes and errors were

systematic within each dataset. We also analyzed the MODIS Global Evapotranspiration Project annual mean actual evaporation, but the values were far from truth.

4. Lack of verification indicators (example: POD, TS, FAR, FBI etc.) in comparison with available rain gauge data or IMD data as reference.

Reply: Thanks for the comment and suggestion. Some of the verification indicators as used in Sikder & Hossain (2016), i.e., POD: the probability of detection. TS: threat Score. FAR: the false alarm ratio. FBI: the frequency bias index, will be used in the revised version. These indexes were used to assess forecasting results with observations. Here we do not have "true observations"- rain gauge data. The IMD dataset will be used as reference although it is interpolated by the India Meteorological Department from rain gauge data. During the period of 1981-2007, the average number of stations per grid point ($1\times1$) varies from 0.2 to 4.4. Fewer stations near the borders of India and in the northern part of the study area and observations are available near the latitude of 35.5N and its north.

Sikder, S., & Hossain, F. (2016). Assessment of the weather research and forecasting model generalized parameterization schemes for advancement of precipitation forecasting in monsoon-driven river basins. *Journal of Advances in Modeling Earth Systems*, *8*(3), 1210–1228. article. http://doi.org/10.1002/2016MS000678

Some specific comments are given below:

As this work is dealing with comparison gridded data sets with varying spatial resolution and other features, the discussion needs to be strengthened incorporating these uncertainty aspects and how reliable/meaningful these absolute precipitation values are, which are used for comparison.

Reply: Thanks for the comment. To achieve this, we added a new section "Comparison with runoff data" under "Discussions". Here we selected three nested catchments in the Beas catchment with respect of different elevation and climate. We indeed tried The Global Runoff Data Centre, but the overlap period of runoff and precipitation is too short. The comparison shows precipitation was underestimated. Though the relationships were different, all datasets were consistent in terms of temporal changes and errors were systematic within each dataset. We also analyzed the MODIS Global Evapotranspiration Project annual mean actual evaporation, but the values were far from truth. For hydrological studies, we recommend the APHRODITE data and the WRF data due matched scales with hydrological processes and considerable spatial variation. Careful local corrections are required.

Need some consistency in acronyms used in this text ( e.g.: authors have used both AGMC and AGCM)

Reply: This is typos. We revised it in the revised version.

Bit more clarity is needed in the description of seasons in the study region (e.g.: in Page 3 authors have considered November-April period as winter, and May to October as summer).

Reply: We apologize for this ambiguity. Here it refers to summer monsoon season and winter monsoon season. The analysis was based on northern meteorological seasons (spring: March to May; summer: June to August; Autumn: September to November; Winter: December to February).

More clarity and justifications through references are required to strengthen "do not suffer from the undercatch problem" explained in the page number 3.

Reply: Thanks. Undercatch means amount of precipitation is not measured by gauges mostly caused by wind turbulence especially for snow. The undercatch leads to a dry biase for unshielded and single gauges at high latitude and high altitude. The undercatch applies to most types of precipitation observations and

their products. Here we meant that Regional climate models simulate precipitation based on mathematical equations and the results are considered not affected by undercatch problem. This part was revised.

Better clarity is needed on the selection of 5 year time slices (1981-1985, 2003-2007) for comparison.

Reply: We will clarify that trend is calculated by the Theil-Sen median method and significance is by the non-parametric Mann-Kendall test, for areal mean of each month and every individual grid of summer and winter. Results showed winter was becoming drier and summer was becoming wetter. However, few (May by the WRF dataset; June by the IMD and ERA-interim datasets, and 10% of grids) of them were statistically significant at the 95% confidence level. As we know, statistic test results are affected by sample size, and we believe the trends will become more significant with time if greenhouse gas emission continues as usual.

Why seasonal comparison is limited to selected two months (Page 6) only?

Reply: The seasonal comparison was shown in Figure 3 whereas these months was purposefully selected to have a better visual impression of spatial distribution and interaction with topography. JA (July and August) and ND (November and December) had respectively highest and lowest precipitation and these months clearly showed the changes.

Page 2 Paragraph 2. This part of text emphases more on to a specific project and associated difficulties- it appears to limit the scope of this work. It would be more appropriate if you define these site-specific difficulties and rationale of this paper through proper references from Himalayan region than describing it as a 'INDICE project' related issue.

Reply: We agree with the reviewer and revised the manuscript accordingly.

I am doubtful about the usefulness of west-to-east latitude based precipitation comparisons for hydrologists in the region. It would be more useful for the hydrologic research community if you could include precipitation PDFs comparisons of larger river basins (Indus River and the upper Ganges River) in this study domain

Reply: This figure aimed to show interactions between atmosphere and topography in JA (July and August, high precipitation months) and ND (November and December, low precipitation months). As shown in the figure, the precipitation changed at different locations and the changes varied among the four datasets. We selected three nested catchments in the Beas catchment with respect of different elevation and climate. We indeed tried The Global Runoff Data Centre, but the overlap period of runoff and precipitation was too short. The comparison showed precipitation was underestimated. Though the relationships were different, all datasets were consistent in terms of temporal changes and errors were systematic within each dataset. We also analyzed the MODIS Global Evapotranspiration Project annual mean actual evaporation, but the values were far from truth. For hydrological studies, we recommend the APHRODITE data and the WRF data due matched scales with hydrological processes and considerable spatial variation.

Titles of Figure 7 and 8: you need to clearly write the details of data points with different colors.

Reply: In Figure 7, the blue color showed the period 1981-1985 and the orange showed the period 2003-2007, as shown in the legend. Figure 8 was removed, because we extended the trend analysis and the significance test to the whole period for every individual grid. Results were shown in Figures 8-12 in the revised version.

---

## Referee Report (RR1)

**Precipitation Pattern in the Western Himalayas revealed by Four Datasets**

**Hong Li, Jan Erik Haugen, Chong-Yu Xu**

HESS manuscript: **hess-2017-296**

Recommendation: **Reconsider after major corrections**

This paper provides a limited examination of four precipitation datasets over the Western Himlayan region of India, and is primarily performed by comparing the spatial pattern and trends of the datasets to identify which datasets are suitable for hydrological modelling. The results are short, but there is a conclusion that WRF and APHRODITE are useful tools. While the paper address an important issue faced by hydro-meteorologists, there are still major problems in the strength and sufficiency of the methodology and of the analyses. While there are improvements in this revised version, there are major issues in the justification of the methodology and the analysis of the results, as well as a lack of detail regarding the WRF modelling hinders the replicability of the results. In my opinion, the current manuscript does not offer sufficiently thorough support to amount to a substantial advancement of understanding precipitation patterns in this region. Therefore, I recommend that the manuscript should be reconsidered for publication after major revisions.

**Major comments:**

1. The major limitation of this manuscript is that there is still a lack of benchmarking to determine which is the "best" dataset for use in hydrological modelling. While the discharge and runoff measurements are a step in this direction, I have to question why the authors do not simply use *in situ* observations as the benchmark, and compare the four data sets to this data. Daily and monthly observed precipitation is freely available from the National Climatic Data Center (https://gis.ncdc.noaa.gov/maps/ncei/cdo/daily) for this region. It would be prudent for this data to be included in this manuscript, given that there is currently no direct benchmark for precipitation used. Although the authors identify the lack of precipitation measurements and their quality is limited in the Himalayan region, including this data and the MODIS data as the benchmark will dramatically improve the veracity of the analysis and conclusions. This will also help determine what added value the "best" dataset has over the others.

2. As well as the spatial pattern and trends in precipitation, whether or not extreme precipitation events are captured by the individual data sets would also be of value to hydro-meteorologists. The "best" data set cannot be identified just from the mean spatio-temporal characteristics, when extreme events are important for accurate hydrological modelling.

3. A missing component of the work as a whole is that there is no proven identification of the real source of the differences. The differences are just explained, without any digging into the simulated processes and forcing data to really find the causes. The key to a valuable comparison study is to at least isolate the source of the differences so that others may understand and build on the work. While the authors identify that differences between the data sets may be attributed to differences in grid size resolution, why not regrid the four data sets onto the same grid and then compare and analyse? On line 26, line 5, the authors state that horizontal resolution is the reason for APHRODITEs better performance. Is this true, or is it an artefact of the individual observations? The authors later attribute ERA-Interim's poor performance to individual observations, rather than horizontal resolution. So which is it, horizontal resolution or an artefact of the number/distribution/quality of observations? This is contradictory and confusing and does not isolate the true reason for differences between data sets.

4. A big question is whether the differences between the data sets are statistically significant. The absence of this is a real deficiency and should be addressed. The reader just doesn't know whether the small differences (e.g. in the spatial differences) have any statistical meaning and which of the different data sets are truly better. It may be that it cannot be concluded that any of these are statistically significantly different. But, even to know that would be of value. Without significance testing results, the reader can't conclude either way.

5. Some very crucial information is required in the WRF section to clarify how the simulations were conducted. It is the only dataset to have been produced by the authors, and the scant detail currently given means that there is no way of replicating their results. Details that need to be clarified in the text are as follows:
    a. What justification is there for using such a precise resolution of 16306 km, rather than 16 km? Maussion *et al.* and Li *et al.* are cited for their model setups, despite both studies using a nested model approach (with the innermost domain less than 3 km). The authors have assumed that model performance (of the other two studies) will scale appropriately to 16306 km. What sensitivity simulations were conducted to test for this? Since this region is surrounded by complex topographic features, and the aim of this paper is to emphasise the benefit of WRF over the other coarser gridded data sets, it would seem appropriate to use a nested model approach in this study going to a resolution of $> 5$km.
    b. The authors state that it is not easy to determine the optimised selection of parameterization schemes – this is correct, and setting the model up for a region often involves intensive testing of the physics options and sensitivity simulations. So please justify why some options from Maussion *et al.* and some options from Li *et al.* have been selected without proper testing? This is contradictory, and there is no adequate justification for doing this (just use one setup or the other).
    c. What was the height of the lowest vertical level (usually ~25m)?
    d. Did the authors consider using other reanalysis products (e.g. MEERA, GFS)?

e. Is there any erroneous data at the boundary edges that run through the Himalayas?

f. What topography data set was used in WRF? Did the authors compare the modelled topography to the real observed topography? Even good elevation agreement, such as being within 10m, can affect the results at the precision reported.

g. At no point in this section have the authors explained how the data has been extracted from WRF. Was the data extracted from the nearest grid point or an interpolation?

h. Were these runs continuous, restarted, or reinitialised, and what was done to check for model spin up and drift? It is really important to make this clear.

i. Where did the SST data come from, and how frequently was this updated?

6. There is still a lack of verification indicators, and including the data from available rain gauges could easily be done. As highlighted above, WMO rain gauge data is freely available. Daily precipitation may also be available from the Bhakra Beas Management Board (see e.g. Norris *et al.* 2017)

Norris, J., Carvalho, L. M., Jones, C., Cannon, F., Bookhagen, B., Palazzi, E., & Tahir, A. A. (2017). The spatiotemporal variability of precipitation over the Himalaya: evaluation of one-year WRF model simulation. *Climate Dynamics*, *49*(5-6), 2179-2204.

**Other comments:**

7. I recommend that the authors carefully proofread the manuscript again. There are numerous grammatical and typographical, none of which I have corrected. Occasionally, the clarity of the arguments is lost by poor sentence structure. This makes the manuscript hard to follow at times.

8. Revise lines 5 – 9 on page 2, the paragraph is hard to follow

9. Although the weaknesses are discussed, I would like to see the advantages of each of the datasets also included in Table 1, as well as in the corresponding paragraph, for a two-sided comparison.

10. Line 5, page 3 – "This interaction brings plenty of precipitation" – This is a qualitative statement which is not useful here. Use the WMO data to state exactly how much; be precise.

11. Lines 16 – 20, page 3 – What is the relevance of this paragraph in this section? What is happening to the glacier? Consider removing, or expand to make relevant.

12. Have you considered also including TRMM (3B42V7) as well as/instead of APHRODITE in the analysis? TRMM is the most reliable decadal dataset of gridded precipitation estimates in the Himalaya (Norris *et al.*), and APHRODITE is mostly based on low elevation sites.

13. Explicitly state where the discharge measurements were sourced. This information appears to be missing.

14. Lines 10 – 12, page 6 – This detail is more appropriate in the method section

15. Lines 16 – 21, page 6 – this paragraph does not make sense at all. Be concise and analytical.
16. Lines 3 – 7, page 7 – This section should also be in the method section. If you are not using *in situ* observations as the benchmark for "best" then use this as your justification in the methodology section.
17. Figure 1 – please show the full WRF domain and river catchments (in panelled plots) so that Figures 2, 6, 8, 10 are more easily understood. You cannot discern ice thickness differences in the figure, so consider removing.
18. Figure 2 – Without a clearer geographical setting, it is difficult to understand what is being shown here. What has been masked? A discrete color bar is needed, as well as a panel of topography.
19. Figure A1 – This is not useful. Include the catchment basins in Figure 1 and remove this figure.

---

## Author Response (AR2)

**Precipitation Pattern in the Western Himalayas revealed by Four Datasets**

**Hong Li, Jan Erik Haugen, Chong-Yu Xu**

HESS manuscript: **hess-2017-296**

Recommendation: **Reconsider after major corrections**

This paper provides a limited examination of four precipitation datasets over the Western Himlayan region of India, and is primarily performed by comparing the spatial pattern and trends of the datasets to identify which datasets are suitable for hydrological modelling. The results are short, but there is a conclusion that WRF and APHRODITE are useful tools. While the paper address an important issue faced by hydro-meteorologists, there are still major problems in the strength and sufficiency of the methodology and of the analyses. While there are improvements in this revised version, there are major issues in the justification of the methodology and the analysis of the results, as well as a lack of detail regarding the WRF modelling hinders the replicability of the results. In my opinion, the current manuscript does not offer sufficiently thorough support to amount to a substantial advancement of understanding precipitation patterns in this region. Therefore, I recommend that the manuscript should be reconsidered for publication after major revisions.

Reply: We first thank the editor and the reviewers for your positive evaluation of our work in general, and we appreciate very much the professional and constructive comments raised by reviewers. In the revised version, we add more information about the WRF model setup. Additionally, we do a Kolmogorov–Smirnov test about difference between four datasets. More importantly, we add data from a rain gauge as a benchmark and compare it with four gridded datasets at various time scales. Last but not least, we carefully revise and improve the manuscript from both scientific presentation as well as English.

**Major comments:**

1. The major limitation of this manuscript is that there is still a lack of benchmarking to determine which is the "best" dataset for use in hydrological modelling. While the discharge and runoff measurements are a step in this direction, I have to question

why the authors do not simply use *in situ* observations as the benchmark, and compare the four data sets to this data. Daily and monthly observed precipitation is freely available from the National Climatic Data Center (https://gis. ncdc. noaa. go v/maps/nce i/cdo/da ily) for this region. It would be prudent for this data to be included in this manuscript, given that there is currently no direct benchmark for precipitation used. Although the authors identify the lack of precipitation measurements and their quality is limited in the Himalayan region, including this data and the MODIS data as the benchmark will dramatically improve the veracity of the analysis and conclusions. This will also help determine what added value the "best" dataset has over the others.

Reply: We thank the reviewer for recognizing our efforts by including MODIS data and for suggestion to compare with the other available dataset. We visit the website of the National Climatic Data Center and indeed find many stations there. We download all precipitation data in the study area for the study period from 1981.01.01 to 2007.12.31. In total, there are 38 stations which have measurements during this period. However, many measurements are missing as shown in Figure A2. The maximum missing rate is as high as 99.99%, and the minimum is 26%, which makes the use of these data somewhat problematic. However, we include *in situ* measurements from a rain gauge, Bhuntar. We get the data from our research partner in India in a previous research project and the data have been used in hydrological modelling for the Beas Basin. More information of this benchmark data and analysis is given in "3.5 Gauge data" and "5.1 Comparison gridded precipitation datasets with gauge data". Of course, we also update other text to make more readable, and all changes are shown in the color of blue.

2. As well as the spatial pattern and trends in precipitation, whether or not extreme precipitation events are captured by the individual data sets would also be of value to hydro-meteorologists. The "best" data set cannot be identified just from the mean spatio-temporal characteristics, when extreme events are important for accurate hydrological modelling.

Reply: Thanks for suggestion. We agree that extreme events are important for accurate hydrological modelling. In the revised version, we include extreme precipitation analysis based on annual maximum daily precipitation. Results show that the WRF dataset gives the best estimations of different quantiles. This part is presented in "5.1 Comparison gridded precipitation datasets with gauge data".

3. A missing component of the work as a whole is that there is no proven identification of the real source of the differences. The differences are just explained, without any digging into the simulated processes and forcing data to really find the causes. The key to a valuable comparison study is to at least isolate the source of the differences so that others may understand and build on the work. While the authors identify that differences between the data sets may be attributed to differences in grid size resolution, why not regrid the four data sets onto the same grid and then compare and analyse? On line 26, line 5, the authors state that horizontal resolution is the reason for APHRODITE's better performance. Is this true, or is it an artefact of the individ ual observations? The authors later attribute ERA-Interim's poor performance to individual observations, rather than horizontal resolution. So which is it, horizontal resolution or an artefact of the number/distribution/quality of

observations? This is contradictory and confusing and does not isolate the true reason for differences between data sets.

Reply: We did not re-grid, as we thought re-gridding coarse resolution to finer resolution only adding more colors rather than really adding information. We have improved the writing for this part. The reasons for differences between datasets are data source and used methodology to produce the data. The spatial resolution is important in showing spatial variability, which should be at a hydrological scale. We state that APHRODITE is much better than IMD, because its resolution is good to more clearly show rain belt at the mountain's foothill. However, we also state that the APHRODITE dataset shows too little precipitation (less than 300 mm/year) at the northeast of the study area, as shown in Figure 2 and "4.1 Spatial variations". All the four datasets are widely used in climate and hydrological studies, but most researchers only select one or two datasets. Here we select four types and they are representative in their own type. We aim to present the differences.

On the other side, we agree that comparing the same spatial resolution is a good way to present the result. Therefore, we interpolate other three datasets to the IMD grids as shown in Figure A1. The general patterns are still the same, but some information is missing at the coarse spatial resolution.

4. A big question is whether the differences between the data sets are statistically significant. The absence of this is a real deficiency and should be addressed. The reader just doesn't know whether the small differences (e.g. in the spatial differences) have any statistical meaning and which of the different data sets are truly better. It may be that it cannot be concluded that any of these are statistically significantly different. But, even to know that would be of value. Without significance testing results, the reader can't conclude either way.

Reply: To address the question of statistical significance, we add Table 3 "P-value of the tailed Kolmogorov–Smirnov test on differences of on annual precipitation (mm/year) among the datasets. The p-value indicates strong evidence against the null hypothesis. It is typically to reject the null hypothesis, which is two datasets are the same here, when p-value is not greater than 0.05." The test is a nonparametric test that can be used to compare a sample with a reference probability distribution (one-sample K–S test), or to compare two samples (two-sample K–S test.

In addition, we include the Bhuntar rain gauge, we not only compare monthly anomaly, but also maximum and minimum, as shown in Figure 12. This is also a type of significance analysis to show how reliable the diffidence is.

5. Some very crucial information is required in the WRF section to clarify how the simulations were conducted. It is the only dataset to have been produced by the authors, and the scant detail currently given means that there is no way of replicating their results. Details that need to be clarified in the text are as follows:
   a. What justification is there for using such a precise resolution of 16306 km,

rather than 16 km? Maussion *et al.* and Li *et al.* are cited for their model setups, despite both studies using a nested model approach (with the innermost domain less than 3 km). The authors have assumed that model performance (of the other two studies) will scale appropriately to 16306 km. What sensitivity simulations were conducted to test for this? Since this region is surrounded by complex topographic features, and the aim of this paper is to emphasise the benefit of WRF over the other coarser gridded data sets, it would seem appropriate to use a nested model approach in this study going to a resolution of $> 5$km.

Reply: The spatial resolution of the WRF dataset is 16 km, and we have corrected the mistake in the manuscript. We have also explained that there are two reasons that we did not use a nested approach. The first one is that the forcing data are at 0.75 degree, and the step from forcing data to the WRF model is only 4. This step is fine in climate simulation. The second reason is consideration for the following hydrological model work. There are several catchments and we are trying to collect more runoff data. It would be too many small nested domains. Therefore, we do not use a nest approach.

b.  The authors state that it is not easy to determine the optimised selection of parameterization schemes – this is correct, and setting the model up for a region often involves intensive testing of the physics options and sensitivity simulations. So please justify why some options from Maussion *et al.* and some options from Li *et al.* have been selected without proper testing? This is contradictory, and there is no adequate justification for doing this (just use one setup or the other).

Reply: There are many options in terms of setting the WRF model. We used a setting that we have used before and it also gave reliable results. We have clarified this part in the revision.

c.  What was the height of the lowest vertical level (usually ~25m)?

Reply: We apologies we did not give the details. The height of the lowest model level depends on the surface pressure. The height of the lowest model level varies between 15 and 27 meters depending on the terrain height. We have added this information in "3.4 WRF dataset" and Table 1.

d.  Did the authors consider using other reanalysis products (e.g. MEERA, GFS)?

Reply: We do not consider other analysis products. We use ERA-interim to force the WRF model and would like to see if the WRF model at high resolution can add more values. As the reviewer mentioned, there are indeed quite many reanalysis products available at present. However, we really cannot include all of them her. Otherwise, it is too much on reanalysis products. We have clarified this part.

e. Is there any erroneous data at the boundary edges that run through the Himalayas?

Reply: We interpret the question as the error at the WRF boundary. The errors are usually shown at three to five grids at the boundary edges. Here we use a very large domain and the study area is a small central area (Figure 1). In addition, the large domain covers a considerable area of ocean, and therefore, the inflow boundary is good.

f. What topography data set was used in WRF? Did the authors compare the modelled topography to the real observed topography? Even good elevation agreement, such as being within 10m, can affect the results at the precision reported.

Reply: We have clarified that we use the 10m topography and land use provided by WRF pre-processing. The source is listed at

http://www2.mmm.ucar.edu/wrf/users/download/get_sources_wps_geog.html ("3.4 WRF dataset"). We agree that elevation is very important and topography plays an important role in precipitation generation mechanism as well as in climate model simulations. We believe that the NCAR and UCAR have compared and concluded that they are of high quality.

g. At no point in this section have the authors explained how the data has been extracted from WRF. Was the data extracted from the nearest grid point or an interpolation?

Reply: We use the native model grids within our study area, and we do not do interpolation except Figure A1. To compare with the Bhuntar rain gauge, we extract data of the nearest point. This description lies in "3.5 Comparison with gauge data" as well as titles of two figures and a table (Figure 11, Figure 12 and Table 5).

h. Were these runs continuous, restarted, or reinitialised, and what was done to check for model spin up and drift? It is really important to make this clear.

Reply: Continuous run of climate models requires too much time and the jobs cannot start in the queue system. Therefore, the model results are restarted about every five years. The simulation is from 1979 and results since 1981 are used here. We have added this description in "3.5 WRF dataset".

We do not mention the drift problem in the manuscript since it usually appears in global climate models. But from the results, the non-parametric Mann-Kendell test for annual precipitation (Table A1), for monthly precipitation (Figure 7), and seasonal precipitation (Figures 8-11 and 13), the WRF simulation does not show additional long term trends compared with other datasets.

6. There is still a lack of verification indicators, and including the data from available rain gauges could easily be done. As highlighted above, WMO rain gauge data is freely available. Daily precipitation may also be available from the Bhakra Beas Management Board (see e.g. Norris *et al*. 2017)

Norris, J., Carvalho, L. M., Jones, C., Cannon, F., Bookhagen, B., Palazzi, E., & Tahir, A. A. (2017). The spatiotemporal variability of precipitation over the Himalaya: evaluation of one-year WRF model simulation. *Climate Dynamics*, *49*(5-6), 2179-2204.

Reply: We checked the website of Bhakra Beas Management Board and did not find any possibility to download data, and there is even no contact information. It is likely that Norris *et al*. 2017 got the data because they have cooperation with their Indian partners. For WMO rain gauge, as mentioned before, we have downloaded and analyzed them, but there are too many missing data and it is difficult to use here. However, we include observation data from a rain gauge, Bhuntar. We get the data from our research partner in India in a previous research project along with the discharge data, and the data have been used in hydrological modelling for the Beas Basin. More information of this benchmark data and analysis is given in "3.5 Gauge data" and "5.1 Comparison gridded precipitation datasets with gauge data". Of course, we also update other text to make more readable, and all changes are marked by blue color.

**Other comments:**

7. I recommend that the authors carefully proofread the manuscript again. There are numerous grammatical and typographical, none of which I have corrected. Occasionally, the clarity of the arguments is lost by poor sentence structure. This makes the manuscript hard to follow at times.

Reply: Thanks for suggestion. We will carefully proofread and correct typo and grammatical errors.

8. Revise lines 5 – 9 on page 2, the paragraph is hard to follow

Reply: They are revised.

9. Although the weaknesses are discussed, I would like to see the advantages of each of the datasets also included in Table 1, as well as in the corresponding paragraph, for a two-sided comparison.

Reply: They are added as the reviewer suggested.

10. Line 5, page 3 – "This interaction brings plenty of precipitation" – This is a qualitative statement which is not useful here. Use the WMO data to state exactly how much; be precise.

Reply: Due to the reason we mentioned before, we refer to India rainfall report 2016 (http://hydro.imd.gov.in/hydrometweb/(S(0rlpja453hfxgw45x041xyqr))/PRODUCTS/Publications/Rainfall%20Statistics%20of%20India%20-%202016/Rainfall%20Statistics%20of%20India%20-%202016.pdf) and revised to be more accurate.

11. Lines 16 – 20, page 3 – What is the relevance of this paragraph in this section? What is happening to the glacier? Consider removing, or expand to make relevant.

Reply: we moved this paragraph to "5.3 Implications for glaciers" and expanded.

12. Have you considered also including TRMM (3B42V7) as well as/instead of APHRODITE in the analysis? TRMM is the most reliable decadal dataset of gridded precipitation estimates in the Himalaya (Norris *et al.*), and APHRODITE is mostly based on low elevation sites.

Reply: We indeed have considered TRMM. However, TRMM is launched in 1997 and there are fewer overlap years with other datasets. Therefore, we do not include her.

13. Explicitly state where the discharge measurements were sourced. This information appears to be missing.

Reply: The discharge stations are operated by Central Water Commission regional office in India. We have added this information in the revised version.

14. Lines 10 – 12, page 6 – This detail is more appropriate in the method section

Reply: They are moved to "3.4 WRF dataset".

15. Lines 16 – 21, page 6 – this paragraph does not make sense at all. Be concise and analytical.

Reply: We removed this in the revised version.

16. Lines 3 – 7, page 7 – This section should also be in the method section. If you are not using *in situ* observations as the benchmark for "best" then use this as your justification in the methodology section.

Reply: They are moved to "3.6 Discharge".

17. Figure 1 – please show the full WRF domain and river catchments (in panelled plots) so that Figures 2, 6, 8, 10 are more easily understood. You cannot discern ice thickness differences in the figure, so consider removing.

Reply: The figure is changed as the reviewer suggestions.

18. Figure 2 – Without a clearer geographical setting, it is difficult to understand what is being shown here. What has been masked? A discrete color bar is needed, as well as a panel of topography.

Reply: The geographical setting shown as the x and y axis. We add elevation and modify figure to be seen more clearly.

19. Figure A1 – This is not useful. Include the catchment basins in Figure 1 and remove this figure.

Reply: Thanks for suggestion. We removed Figure A1 and revised Figure 1. The catchment basins are very small compared to the whole study area. In the new version, we add also the boundary of India and disputed area as well as small window map to show their locations more clearly.

**Reviewer 2**
Figure 1: This figure is not following proper international boundaries. Wrong depiction of map of India (shaded region)

Reply:  We are sorry for disputed boarders and we change the label to "India and disputed area". Figure A1: Very small map, and geographical location of Beas is not clear from this map.

Reply:  Thanks for suggestion. We removed Figure A1 and revised Figure 1. The catchment basins are very small compared to the whole study area. In the new version, we add also the boundary of India and disputed area as well as small window map to show their locations more clearly.